

# Atmospheric gaseous hydrochloric and hydrobromic acid in urban Beijing, China: detection, source identification and potential atmospheric impacts

Xiaolong Fan[1#], Jing Cai[2#], Chao Yan[2], Jian Zhao[2], Yishuo Guo[1], Chang Li[1], Kaspar R. Dällenbach[2], Feixue Zheng[1], Zhuohui Lin[1], Biwu Chu[3,4], Yonghong Wang[2], Lubna Dada[2], Qiaozhi Zha[2], Wei Du[2], Jenni Kontkanen[2], Theo Kurtén[5], Siddhart Iyer[6], Joni T. Kujansuu[1,2], Tuukka Petäjä[2], Douglas R. Worsnop[7], Veli-Matti Kerminen[2], Yongchun Liu[1], Federico Bianchi[2], Yee Jun Tham[2*], Lei Yao[2*], Markku Kulmala[1,2,8]

**Affiliations:**

[1]Aerosol and Haze Laboratory, Beijing Advanced Innovation Center for Soft Matter Science and Engineering, Beijing University of Chemical Technology, Beijing 100089, China

[2] Institute for Atmospheric and Earth System Research / Physics, Faculty of Science, University of Helsinki 00560, Finland

[3] State Key Joint Laboratory of Environment Simulation and Pollution Control, Research Center for Eco-Environmental Sciences, Chinese Academy of Sciences, Beijing 100085, China

[4] Center for Excellence in Regional Atmospheric Environment, Institute of Urban Environment, Chinese Academy of Sciences, Xiamen 361021, China

[5] Department of Chemistry, University of Helsinki, FI-00014 Helsinki, Finland

[6] Aerosol Physics Laboratory, Physics Unit, Tampere University, Tampere 33100, Finland

[7] Aerodyne Research Inc., Billerica, Massachusetts 01821, USA

[8] Joint International Research Laboratory of Atmospheric and Earth System Sciences (JirLATEST), Nanjing University, Nanjing 210023, China.

[#] These authors contributed equally.

**Correspondence**: Lei Yao (lei.yao@helsinki.fi) and Yee Jun Tham (yee.tham@helsinki.fi)





**Abstract**


Gaseous hydrochloric (HCl) and hydrobromic acid (HBr) are vital halogen species that play essential roles in tropospheric physicochemical processes. Yet, majority of the current studies on these halogen species were conducted in marine or coastal areas. Detection and source identification of HCl and HBr in inland urban areas remain scarce, thus, limiting the full understanding of halogen chemistry and potential atmospheric


impacts in the environments with limited influence from the marine sources. Here, both gaseous HCl and HBr were concurrently measured in urban Beijing, China during winter and early spring of 2019. We observed significant HCl and HBr concentrations ranged from a minimum value at $1.3\times10^8$ cm$^{-3}$ and $4.3\times10^7$ cm$^{-3}$ up to $5.9\times10^9$ cm$^{-3}$ and $1.2\times10^9$ cm$^{-3}$, respectively. The HCl and HBr concentrations are enhanced along with the increase of atmospheric temperature, UVB, and levels of gaseous HNO$_3$. Based on the air mass


analysis and high correlations of HCl and HBr with the burning indicators (HCN and HCNO), the gaseous HCl and HBr are found to be related to anthropogenic burning aerosols. The gas-aerosol partitioning may also play a dominant role in the elevated daytime HCl and HBr. During the daytime, the reaction of HCl and HBr with OH radicals lead to significant production of atomic Cl and Br, up to $1.7\times10^4$ cm$^{-3}$ s$^{-1}$ and $7.9\times10^4$ cm$^{-3}$ s$^{-1}$, respectively. The production rate of atomic Br (via HBr + OH) are 2-3 times higher than that of


atomic Cl (via HCl + OH), highlighting the potential importance of bromine chemistry in the urban area. Furthermore, our observations of elevated HCl and HBr may suggest an important recycling pathway of halogen species in inland megacities, and may provide a plausible explanation for the widespread of halogen chemistry, which could affect the atmospheric oxidation in China.












## 1. Introduction

Tropospheric halogen chemistry plays a variety of roles in perturbing the fate of chemical compositions, including ozone ($O_3$) and volatile organic compounds (VOCs) in the troposphere (Saiz-Lopez and von Glasow, 2012;Simpson et al., 2015; Artiglia et al., 2017). Halogen radicals, in particular the atomic chlorine (Cl·) and bromine (Br·), can deplete the $O_3$, and react rapidly with VOCs with reaction rates up to two orders of magnitude faster than that of the hydroxyl radical (OH) reaction with VOCs (Atkinson et al., 2007). Significant halogen-induced $O_3$ reduction of about 10% of the annually averaged tropospheric ozone column was reported over the tropical marine boundary layer (Saiz-Lopez et al., 2012). However, in polluted coastal regions with high $NO_x$, the coupling between halogen chemistry and $NO_x$ chemistry contributes to significant enhancement of ozone production of up to 7 ppb (parts per billion by volume) (Li et al., 2020;Sherwen et al., 2017;Sarwar et al., 2014). Besides affecting the ozone chemistry, the oxidation processes of VOCs by halogen radicals can potentially lead to secondary aerosols production. Wang and Hildebrandt Ruiz (2017) demonstrated that the chlorine-initiated oxidation of isoprene contributed to the formation of particulate organochloride and the yield of secondary organic aerosol (SOA) ranged from 7 to 36% (Wang and Ruiz, 2017). A recent study also found that the oxidation of alpha-pinene by chlorine atoms yields low-volatility organic compounds, which are essential precursors for secondary particle formation and growth (Wang et al., 2020).

It is known that sea-salt particle is a major source of atomic halogens in the marine environment. The chloride ($Cl^-$) and bromide ($Br^-$) in the sea-salt particles can be displaced by strong acids (i.e. nitric acid, $HNO_3$) to release gas-phase hydrogen halides HX (reaction (R1); X = Cl or Br) into the atmosphere (Gard et al., 1998). The HX then can react with an OH radical to form a X· via reaction (R2).

$$X^- \text{ (sea-salt)} + HNO_3 \rightarrow HX \qquad (R1)$$

$$HX + OH \rightarrow X\cdot + H_2O \qquad (R2)$$

On the other hand, the heterogeneous uptake of dinitrogen pentoxide ($N_2O_5$) onto sea-salt particles can form nitryl halides $XNO_2$ via reaction (R3) (Finlayson-Pitts et al., 1989;Osthoff et al., 2008;Tham et al., 2014), which is a reservoir of halogen during the nighttime. At sunrise, the $XNO_2$ undergoes rapid photolysis to liberate highly reactive halogen atom (X·), which subsequently reacts with VOCs to produce HX and peroxy radicals ($RO_2$; reaction (R4) and (R5)). Besides, the heterogeneous oxidation of $Br^-$ by $O_3$ at the aqueous phase-vapour interface can lead to the formation of a pre-complex intermediate ($Br \cdot OOO^-$) which contributes the formation of atmospheric HOBr (Artiglia et al., 2017).

$$X^- \text{ (sea-salt)} + N_2O_5 \rightarrow XNO_2 + NO_3^- \qquad (R3)$$

$$XNO_2 + h\nu \rightarrow X\cdot + NO_2 \qquad (R4)$$

$$X\cdot + RH \rightarrow HX + RO_2 \qquad (R5)$$

The atmospheric lifetimes of HCl and HBr due to reaction (R2) are approximately 35.6 h and 2.5 h (when OH = $1 \times 10^7$ molecules $cm^{-3}$), respectively, making them a significant daytime recycling source of atomic halogen in the marine atmosphere. Riedel et al. (2012) showed that the reaction of HCl with OH accounts for about 45% of the integrated Cl atom production over the entire day along the Santa Monica Bay of Los Angeles (Riedel et al., 2012). Another ship-borne study reported that the Cl atom production rate peaks at $2.9 \times 10^5$ molecules $cm^{-3}$ $s^{-1}$ during the noontime in Southern Coastal California (Crisp et al., 2014). The produced HCl and HBr can also end up in particle phase during the nighttime (Chen et al., 2016;Roberts et al., 2019;Crisp et al., 2014), and further promoting the heterogeneous reaction of $N_2O_5$ (R3).

The discovery of Thornton et al. has changed the paradigm of halogen chemistry, where it was thought to be restricted to the marine environment (Thornton et al., 2010). A significant source of atomic chlorine from


the heterogeneous reaction of $N_2O_5$ onto chloride aerosol (R3) was observed in Boulder, U.S., which is 1400 km from the nearest coastline, indicating that active chlorine chemistry also occurs in the region far from the ocean (Thornton et al., 2010). Follow-up studies have confirmed the presence of halogen activation spreading over the continental regions of North America, Europe and Asia (Mielke et al., 2011;Phillips et al., 2012;Riedel et al., 2013;Tham et al., 2016;Wang et al., 2017;Tham et al., 2018;Liu et al., 2017;Xia et al., 2020;Zhou et al., 2018). These findings suggest a crucial role for HCl gas-particle partitioning in sustaining

the aerosol chloride concentrations in continental regions for reaction (R3) to take place (Brown and Stutz, 2012).

On the global scale, sea salt sprays were estimated to be the dominant source of halogens such as Cl and Br (Wang et al., 2019;Keene et al., 1999). Through acid displacement and other heterogeneous processes, 64 Tg $a^{-1}$ and 6.2 Tg $a^{-1}$ gas-phase inorganic Cl and Br from sea salt were emitted to the troposphere, while

anthropogenic emissions such as biomass burning, fossil combustion and incineration were supposed to be minor on a global scale (Wang et al., 2019;Keene et al., 1999). For the emissions of Cl, anthropogenic emissions were quite crucial for both gaseous and particulate Cl in the urban environment and heavily polluted areas. For example, the anthropogenic emissions for gaseous HCl and particulate Cl were 458 and 486 Gg in 2014 in China, of which biomass burning is the largest contributor (Fu et al., 2018a). Many recent

field studies reported elevated $ClNO_2$ and particulate chloride concentrations in the plumes influenced by biomass burning and coal-fired power plants, suggesting they could be the driving force for the Cl activation process in continental areas (Riedel et al., 2013;Tham et al., 2016;Wang et al., 2017;Liu et al., 2017;Yang et al., 2018). Furthermore, Bannan et al.(2019) showed that the $ClNO_2$ is consistently formed at a landfill site in London, highlighting the potential contribution from landfill emissions of Cl in promoting the reactions

(R3) and (R4) (Bannan et al., 2019). Other possible anthropogenic Cl sources include the emissions from industrial, and water and sewage treatment plants (Hara et al., 1989;Graedel and Keene, 1995;Thornton et al., 2010). During the wintertime, the use of road salt could also be a dominant source of atmospheric Cl in the city areas (McNamara et al., 2020).

The atmospheric bromine is much less abundant than chlorine in the troposphere with the concentrations

of around 25 ppt (parts per trillion by volume) compared to 3.7 ppb of chlorine (Bedjanian and Poulet, 2003). HBr is known as a principal bromine sink species for the ozone loss chemistry in the stratosphere showing the average concentration of 1.31±0.39 ppt between 20.0 to 36.5 km altitude (Bedjanian and Poulet, 2003;Nolt et al., 1997;Yang et al., 2005), and also one of the dominant inorganic bromine species in the marine boundary layer, free troposphere and tropical tropopause layer as well (Fernandez et al., 2014;Glasow

and Crutzen, 2014;Nolt et al., 1997;Bedjanian and Poulet, 2003). In the urban environment, atmospheric Br was previously known to be strongly affected by traffic emissions since ethylene dibromide ($C_2H_4Br_2$) was used to be as anti-knock compounds to leaded gasoline (Glasow and Crutzen, 2014). Yet, since the phasing out of leaded gasoline, the long-term atmospheric Br exhibited a continuous decreasing trend for 2 to 3 decades atmospheric Br in Germany (Lammel et al., 2002), and a similar situation is expected in Beijing as

the usage of leaded gasoline was banned from the years around the 2000s in China (Cai et al., 2017).

Despite the advances in the understanding of concentrations and sources of global halogen species, the atmospheric gaseous HCl and HBr in the continental, especially urban environments, are much less studied. Some limited studies focused on the atmospheric HCl, for example, Crisp et al. (Crisp et al., 2014) summarized that the concentration of HCl is typically less than 1 ppb over the continental regions, while an

airborne measurement showed HCl concentrations of around 100 ppt was typically observed over the land area of northeast United States, except near power plant plumes with the concentrations over 1 ppb (Haskins et al., 2018). Furthermore, much less information is available on the presence of HBr in the continental



environment. Until very recently, an airborne measurement detected significant levels of gas-phase reactive bromine species in the exhaust of coal-fired power plants (Lee et al., 2018). Therefore, the measurement of

gas-phase HCl and HBr in inland urban environments is of necessary to fully assess their effects on the tropospheric chemistry, such as gas-particle partitioning effects on the particulate halide concentrations that can undergo rapid activation via reaction (R3). Those would be more important in polluted regions such as the North China Plain, where Beijing is located in and a large amount of chloride were emitted to the atmosphere (Tham et al., 2016;Zhou et al., 2018;Fu et al., 2018b).

In this study, we deployed a Chemical Ionization-Atmospheric Pressure interface-long-Time-Of-Flight mass spectrometer (CI-APi-LTOF) to measure the atmospheric gas-phase HCl and HBr from February 1 to March 31 2019, in urban Beijing, China. To our best knowledge, it is the first time presenting a simultaneous measurement of HCl and HBr with high time-resolution in urban Beijing. Besides, we identify the potential source that contributed to the high levels of gaseous HCl and HBr during wintertime and early springtime. In

addition, we estimate the contribution of gaseous HCl and HBr on the production rates of atomic Cl and Br in urban Beijing.

## 2. Methodology

### 2.1 Sampling site.

The field measurements were conducted at Beijing University of Chemical Technology (BUCT)

monitoring station (39.94° N, 116.30° E), located in an urban area of Beijing, China (Figure 1) where the nearest coastline locates about 150 km away in the southeast. The sampling site is about 130 m north to the Zizhuyuan Road and 550m west to the West Third Ring Road, which is one of the main roads in Beijing. Besides the effect of traffic, this site is also surrounded by local commercial properties and residential dwellings. Thus, the BUCT sampling site can be regarded as a typical urban site. More information about

this sampling site can be found in previous studies (Cai et al., 2020;Kontkanen et al., 2020;Zhou et al., 2020;Chu et al., 2021). The instruments were deployed at the roof of a teaching building, which is approximately 15 m above the ground level.

### 2.2 CI-APi-LTOF mass spectrometer.

The working principle of CI-APi-LTOF (Aerodyne Research Inc. and Tofwerk AG) has been described

elsewhere (Yao et al., 2020;Eisele and Tanner, 1993;Yao et al., 2018), therefore only details relevant to this present work were discussed. A typical mass spectrum during our field measurement was depicted in Figure S1. The dominant reagent ions were nitrate ions ($NO_3^-$, and $HNO_3 \cdot NO_3^-$) and nitrite ions ($NO_2^-$). Among them, nitrate ions were generated by exposure of sheath flow (pure air with RH ~5%) which carried gaseous $HNO_3$. Besides the nitrate ions that acted as dominate reagent ions, nitrite ions were formed from the reaction of a

small amount of $NO_2$ (~1 ppb) in the sheath flow with $O_2^-$ and $OH^-$ which were generated from the exposure of sheath flow (pure air with RH ~5%) to an X-ray source (Hamamatsu L9491) (Figure S5) (Arnold et al., 1995;Skalny et al., 2004). Considering nitrate ions were still the dominant reagent ions (Figure S1), the CI-APi-LTOF was actually operated as a typical nitrate-CI-APi-LTOF.

Ambient air was drawn into the CI-inlet through a 3 quarter-inch stainless steel tube with a flow of ~8 L

$min^{-1}$. A small mixed flow (~0.8 L $min^{-1}$ controlled by a critical orifice with 300 μm diameter) entered the APi-LTOF and be analyzed. The CI-APi-LTOF was operated in the negative V-mode with the mass resolving power of ~10000 Th/Th and the mass accuracy better than 5 ppm. Data of CI-APi-LTOF were acquired with 5 s time resolution, and the recorded data were further analyzed with a MATLAB tofTools package (Junninen et al., 2010).





### 2.3 Detection and quantification of HCl and HBr

From Table 1, the Gas-phase acidity ($-\Delta G$) of HCl is 1354 kJ mol$^{-1}$ which is larger than that of HNO$_3$ (1329 kJ mol$^{-1}$). Besides, the enthalpy ($\Delta H$) of HNO$_3$ and Cl$^-$ is 32.8 kcal mol$^{-1}$, which is higher than that of HCl and NO$_3^-$ (22.9 kcal mol$^{-1}$) hinting that the reaction of HCl and NO$_3^-$ was unlikely to occur (Figure S4a). Additionally, from a previous study, the reaction rate ($< 10^{-12}$ cm$^{-3}$ s$^{-1}$) between NO$_3^-$ and HCl was significantly less than that ($1.4\times10^{-9}$ cm$^{-3}$ s$^{-1}$) of NO$_2^-$ with HCl (Ferguson et al., 1972). Therefore, the HCl is likely mainly charged by NO$_2^-$ instead of NO$_3^-$ to result in Cl$^-$ formation. The ion-molecule reaction between nitrite ions and HCl can be written as follows (Ferguson et al., 1972):

$$NO_2^- + HCl \rightarrow Cl^- + HNO_2 \qquad (R6)$$

In addition to NO$_2^-$, the HCl can also react with O$_2^-$, leading to Cl$^-$ and Br$^-$ formation via reaction (R7).

$$O_2^- + HCl \rightarrow Cl^- + HO_2 \qquad (R7)$$

Therefore, HCl can be quantified according to:

$$[HCl] = C_{HCl} \times \frac{[Cl^-]}{(NO_2^- + O_2^-)} \qquad E\ (1)$$

where $C_{HCl}$ (in units of cm$^{-3}$) is a calibration coefficient of HCl. Based on ambient data, a very small fraction (less than 5%) of Cl$^-$ (or HCl) would react with HNO$_3$ (or NO$_3^-$) in the sheath flow to form Cl$^-\cdot$HNO$_3$ (or HCl$\cdot$NO$_3^-$). Thus, the signals of Cl$^-\cdot$HNO$_3$ (or HCl$\cdot$NO$_3^-$) were not taken into account for HCl quantification. Using 4-days synchronous gaseous HCl concentrations measured by a Monitor for AeRosols and Gases in Ambient air (MARGA, Metrohm Inc., Switzerland), an indirect calibration was adopted to quantify the HCl measured by the CI-APi-LTOF (Section S5 in Supporting Information). The obtained calibration factor $C_{HCl}$ for HCl is $2.84\pm0.07\times10^{12}$ cm$^{-3}$ (Figure S8b).

On the basis of $-\Delta G$ of HBr, HNO$_3$, HNO$_2$ and HO$_2$ and the enthalpy ($\Delta H$) calculations (Table 1, Figure 2 and S4), besides the reaction with NO$_2^-$ and O$_2^-$, similar with HCl, some of HBr could also react with NO$_3^-$ to form Br$^-$ via the reaction (R8) (Ferguson et al., 1972).

$$NO_3^- + HBr \rightarrow X^- + HNO_3 \qquad (R8)$$

Hence, the HBr should be quantified according to:

$$[HBr] = C_{HBr} \times \frac{[Br^-]}{(NO_2^- + O_2^-) + (NO_3^-)} \qquad E\ (2)$$

where $C_{HBr}$ (in units of cm$^{-3}$) is a calibration coefficient of HBr. However, due to a direct calibration for HBr was not available, the calibration coefficient of HCl ($C_{HCl}$) was utilized to semi-quantify HBr based on the following equation:

$$[HBr] = C_{HCl} \times \frac{[Br^-]}{(NO_2^- + O_2^-)} \qquad E\ (3)$$

Since the enthalpies ($\Delta H$) of HBr$\cdot$NO$_3^-$ formed by HBr with NO$_3^-$ (27.3 kcal mol$^{-1}$) and Br$^-$ with HNO$_3$ (27.9 kcal mol$^{-1}$) were very close to each other (Figure S4b), it was difficult to quantify the specific contribution to Br$^-$ from the reaction of HBr with NO$_3^-$. Also, the ratios of Br$^-\cdot$HNO$_3$ (or HBr$\cdot$NO$_3^-$) to Br$^-$ were less than 4%. Therefore, in the equation 3, the reaction pathway of HBr with NO$_3^-$ was not considered. The presented HBr concentrations should be treated as semi-quantification ones and upper limit values.

To confirm these ion-molecule reactions, high concentrations (undetermined) of gaseous HCl and HBr were mixed with zero air generated from a zero-air generator (Aadco 737), and then measured by CI-APi-LTOF (Section S4). After the injection of HCl and HBr, the signals of Cl$^-$, Br$^-$, Cl$^-\cdot$HNO$_3$ (or HCl$\cdot$NO$_3^-$) and



Br$^-$·HNO$_3$ (or HBr·NO$_3^-$) started to increase (Figure S7), confirming that the HCl and HBr can be detected as Cl$^-$, Br$^-$, Cl$^-$·HNO$_3$ and Br$^-$·HNO$_3$ by CI-APi-LTOF.


### 2.4 Other auxiliary measurements.

Gaseous HCN and HCNO also can be detected by O$_2^-$ through the ion-molecule reactions as follows:

$$O_2^- + HCN \rightarrow CN^- + HO_2 \qquad (R9)$$

$$O_2^- + HCNO \rightarrow CNO^- + HO_2 \qquad (R10)$$

The -$\Delta G$ of HCN and HCNO are 1433 kJ mol$^{-1}$ and 1415 kJ mol$^{-1}$, respectively, which are higher than that of NO$_2^-$ (1393 kJ mol$^{-1}$) (Table 1), and lower than that of O$_2^-$ (1450 kJ mol$^{-1}$). Therefore, HCN and HCNO are able to be charged by O$_2^-$ (but not NO$_2^-$) via deprotonation reaction to lead to CN$^-$ and CNO$^-$ formation. In this study, direct calibrations for HCN and HCNO were not available. Instead, the normalized signals of CN$^-$ and CNO$^-$ by O$_2^-$ were tentatively utilized to indicate the abundance and trend of HCN and HCNO.

The meteorological parameters, including temperature and UVB intensities, were recorded by a weather station (Vaisala Inc., Finland). The mass concentrations of particulate chlorine and black carbon (BC) in PM$_{2.5}$ were measured by a time-of-flight aerosol chemical speciation monitor (ToF-ACSM, Aerodyne Research Inc., USA) and an aethalometer (AE33, Magee Inc., USA), respectively (Section S1 in Supporting Information).

### 3.    Results and discussions

### 3.1 HCl and HBr measurement.

Figure 3 shows the time series of gaseous HCl and HBr, temperature (T), and ultraviolet radiation b (UVB, 280-315 nm) intensities for the entire measurement period in winter and early spring of 2019 (February to April). High concentrations of HCl and HBr were observed for the whole measurement period with a clear diurnal variation (Figure 3c). The mean concentrations of HCl and HBr are $1.3 \pm 1.1 \times 10^9$ and $1.9 \pm 1.5 \times 10^8$ molecules cm$^{-3}$, respectively. The maximum concentrations reach up to $5.9 \times 10^9$ cm$^{-3}$ for HCl, and $1.2 \times 10^9$ cm$^{-3}$ for HBr during the daytime. The concentrations of HCl and HBr showed a similar change in atmospheric temperature and UVB. For the first period of measurement (from February 1 to February 15), HCl and HBr concentrations are lower when the atmospheric temperature is close to 0°C and UVB intensities are relatively low, while the HCl and HBr concentrations begin to increase together with the rising of temperature and UVB during April 2019.

The diurnal cycles of HCl and HBr are depicted in Figure 4a and 4b, respectively. The HCl concentrations are typically higher than HBr by approximately an order of magnitude; nevertheless, the diel patterns showed by these two species are quite similar to each other. It is noticed that both HCl and HBr began to increase after sunrise, and relatively high concentration was observed during the daytime (8:00 to 17:00). Our observation of daily averaged mass concentrations of particulate chloride (Cl) in PM$_{2.5}$ showed a similar trend with daily averaged gaseous HCl (Figure S9a). In contrast, the diurnal variations of HCl and particulate Cl showed the opposite trend at daytime from 08:00 to 17:00 (Figure S9b). The ratios of gaseous HCl to particulate Cl ranged from ~0.1 at nighttime and early morning to ~0.3 around noontime (Figure S9c), implying that there is intense gas to particle partitioning during the daytime. It also can be found that elevated temperature and high abundance of HNO$_3$ could intensify the gas to particle partitioning in the daytime (Figure 4d). This is consistent with our observation above where the increase of temperature and UVB could reinforce the formation of chemicals (e.g., HNO$_3$) that promote the gas-aerosol partitioning or directly increase gas-phase formation rate of HCl and HBr (Crisp et al., 2014;Riedel et al., 2012), thus further enhancing the HCl and HBr. Although there is no direct measurement of particulate bromide (Br), considering





the diurnal pattern of HBr and the good correlation ($r = 0.70$) between HBr and HCl (Figure 4c), it is rational to suppose HBr also derived from gas-aerosol partitioning process.

These observations showed that there is an abundance of gaseous HCl and HBr in the polluted urban environment. To our best of knowledge, this is the first concurrent observation of gaseous HCl and HBr in a polluted inland urban atmosphere. Although it is well known that the HCl is abundant in the polluted coastal regions, previous studies show that the typical HCl mixing ratios over the continental urban areas are less than 1 ppb (Crisp et al., 2014;Faxon and Allen, 2013;Le Breton et al., 2018), which are similar to our observations at Beijing. In contrast, the presence of gaseous HBr in the urban regions is unknown prior to our observation. The significant concentration of HBr in the urban atmosphere of Beijing is even comparable to the simulated concentrations in the marine environment, where concentration up to 2 ppt was reported (Fernandez et al., 2014). These elevated HCl and HBr in the urban of Beijing may point to the existence of Cl and Br sources in this region.

**3.2 Source identification.**

The natural sources of atmospheric Cl and Br include sea salt spray, wildfires and volcano emissions, while the anthropogenic emissions include coal combustions, traffic emissions as well as other industries such as pesticides, battery industry and waste incinerations (Simpson et al., 2015). Comparing with the sources of particulate Cl and Br that are widely studied and identified in previous literature, the origins of gaseous HCl and HBr are much less studied, due to their much shorter lifetime in the troposphere (Simpson et al., 2015).

According to air mass analysis (24h back-trajectory) for HCl and HBr during February and March (Figure 5a and b), the potential source regions of the selected periods with high-level concentrations of HCl (above 75% percentile) were located in the south of North China Plain, such as the south of Hebei province where heavy residential coal, biomass burning and industries emissions occurred (Fu et al., 2018b). Those figures further suggested that the high concentrations of HCl seemed not to be strongly affected by marine regions during our sampling period. Instead, the good correlation ($r = 0.67$) between hourly particulate Cl and BC together with the similar trend between particulate Cl and HCl suggested that HCl is likely to have the same original sources with particulate Cl and black carbon (BC) in $PM_{2.5}$ rather than marine sources (Figure S9a and Figure S10a). Hydrocyanic acid (HCN) and isocyanic acid (HCNO), which were typically regarded as tracers for burning emissions, especially biomass burning process (Vigouroux et al., 2012;Adachi et al., 2019;Leslie et al., 2019;Wren et al., 2018;Priestley et al., 2018). Although a recent study showed that HCNO came from both primary emissions and secondary formation in the scale of North China Plain (NCP) during the daytime (Wang et al., 2020), the high correlations between HCN and HCNO (daytime, 08:00-17:00, $r = 0.94$ and nighttime, 18:00-07:00, $r = 0.96$) indicated that in urban Beijing, HCN and HCNO are mainly from primary emission (Figure 6c) and can be regarded as the tracers of combustion emissions. Thus, high correlations of measured gaseous HCl with HCN ($r = 0.83$) and HCNO ($r = 0.90$) further suggested that the HCl during our sampling period was more likely coming from combustion origins rather than marine source in the urban Beijing (Figure 6a and b). Since gaseous HCl could be affected by both emissions and gas/particle partitioning (shown in Figure 4d), we compared the daily concentrations of gaseous HCl and particulate Cl to minimize the influence of temperature and partitioning. The daily averaged HCl concentration had a high correlation with daily averaged particulate Cl ($r = 0.84$ and 0.70 for winter and spring periods, respectively) and BC concentration ($r = 0.82$), which is consistent with previous studies that particulate Cl, coal combustion organic aerosol (CCOA) and BC were highly correlated and likely to be from the same source in winter of Beijing (Hu et al., 2017;Hu et al., 2016).

Similar to HCl, the potential source regions for high Br concentrations were also located in the inland,





demonstrating marine sources might not be the dominant source for gaseous HBr in winter of Beijing (Figure 5b). The ratio of particulate Br/Na from previous literature in Beijing was 0.04 (He et al., 2001), which was much higher than those from seawater (0.018) and crustal dust (0.0006 to 0.0008) but much closer to those of biomass burning aerosols (0.01 to 0.06) (Sander et al., 2003). As discussed before, the good correlation ($r$ = 0.70) between gaseous HCl and HBr also implied that their similar origins. In our study, moderate

correlation coefficients were also observed between gaseous HBr and combustion tracers such as HCN, HCNO (0.63 and 0.62, respectively) and daily BC ($r$ = 0.60) (Figure 6a, 6b and S10b). Multiple gaseous organic and inorganic Br compounds such as $CH_3Br$, $Br_2$, $BrNO_2$, $BrCl$, $CH_3Br$ and $CH_2Br_2$ were also observed in different combustion processes such as biomass burning, coal combustions and waste incineration in previous studies, further supporting the possibilities of combustion origins of the gaseous HBr

in this study (Lee et al., 2018;Keene et al., 1999;Manö and Andreae, 1994). A recent airborne observation conducted in the U.S. found that high levels of reactive inorganic Br species in the plume from a coal power plant, likely due to the application of calcium bromide as additives in coal fuel (Lee et al., 2018). Together all these, in urban Beijing, the measured HBr was more likely coming from combustion sources such as biomass burning and coal combustion in the south of Beijing rather than marine sources. It is also interesting

to note that in a previous study, gaseous Br was found to be 4 to 10 times higher than particulate Br (Moyers and Duce, 1972). In addition, from previous observations, gaseous organic bromine was around 7 ng m$^{-3}$ in Beijing, of which 6 ng m$^{-3}$ was extractable and able to release to the atmosphere (Tian et al., 2005). Considering the high concentration and reactivity of both organic/inorganic Br, gaseous Br from anthropogenic sources may play a more critical role in the urban atmosphere.

**3.3 Halogens atom production.**

To investigate the potential atmospheric implications of HCl and HBr on atmospheric oxidation capacity, we calculated the production rate of atomic Cl ($P_{Cl\cdot}$) and Br ($P_{Br\cdot}$) via the reactions of HCl and HBr with OH radicals. Figure 7 shows the time series of $P_{Cl\cdot}$, $P_{Br\cdot}$, and the estimated diel concentration of OH calculated from photolysis rate ($J_{O1D}$ and $J_{NO_2}$) and $NO_2$ concentration ($C_{NO_2}$) (Section S8). Note that the estimated peak

concentrations of OH radicals varied between ~2.8×10$^5$ to ~4.3×10$^6$ molecules cm$^{-3}$ during noontime. The reaction of HCl with OH radicals lead to a daily mean Cl atom production rate of 3.0×10$^3$ molecules cm$^{-3}$ s$^{-1}$ (Figure 7b). These rates fall within the range of Cl atom production rates (~10$^3$ to 10$^6$ molecules cm$^{-3}$ s$^{-1}$) reported in polluted environments (Crisp et al., 2014;Hoffmann et al., 2018). For the reaction of HBr with OH, it is estimated to produce a daily mean of 8.4×10$^3$ molecules cm$^{-3}$ s$^{-1}$ of Br atom (Figure 7b). This result

shows that in addition to the Cl atom, Br atom could also be present in urban Beijing and may act as important as the Cl atom in term of reaction with OH, since the $P_{Br\cdot}$ is about 2-3 times faster than the $P_{Cl\cdot}$. (Figure 7b). Recent studies in several polluted sites of China suggested that the photolysis of $ClNO_2$ and $Cl_2$ are the dominant daytime Cl atom sources, while the reaction of HCl with OH may also act as important recycling of Cl atom, which ultimately enhanced the atmospheric oxidation capacity (Tham et al., 2016;Liu et al.,

2017;Xia et al., 2020). In analogous to the chlorine chemistry, the reaction of HBr with OH could contribute to the recycling of Br atom, on top of the significant production from rapid photolysis of $Br_2$ and $BrNO_2$, which are likely ubiquitous in a polluted urban environment since high levels of $Br_2$ and $BrNO_2$ were measured in the coal-fired power plant plumes (Lee et al., 2018).

**4. Conclusions**

In conclusion, we present the first concurrent measurement of both gaseous HCl and HBr in urban Beijing, a megacity with strong anthropogenic emissions in the North China Plain. Our observation surprisingly shows significant concentrations of HBr in urban Beijing, together with the elevated levels of HCl, throughout the winter and spring during our sampling period. Gaseous HCl and HBr are most likely originated from



anthropogenic emissions such as burning activities (e.g., biomass burning and fossil fuel combustion) in the inland region rather than marine sources. Besides, the gas-aerosol partitioning may play a crucial role in contributing to elevated levels of HCl and HBr in urban Beijing. These abundant HCl and HBr in the polluted urban troposphere may further influence the photochemistry of the atmosphere through the following two aspects: (1) direct contributions to the production of highly reactive halogen atom (e.g., Cl· and Br·), which can rapidly oxidize VOCs (reaction (R5)); (2) replenishing the halide ion (Cl$^-$ and Br$^-$) in the aerosols for supporting the nocturnal heterogeneous production of $ClNO_2$ and $BrNO_2$, major sources of highly reactive halogen atom at sunrise (reaction (R3) and (R4)). Our observation of elevated HCl and HBr may indicate an important recycling pathway of Cl and Br species, and may provide a plausible explanation to the recent observations of widespread halogen activation in polluted areas of China (e.g. Tham et al., 2016;Zhou et al., 2018;Xia et al., 2020), which could have a significant influence on the atmospheric oxidation capacity and secondary aerosol formation. Furthermore, the additional insight on the HBr levels at Beijing shows that the bromine chemistry, a previously neglected chemistry, may be important in inland megacities of China. Our results also suggest that understanding of gaseous HCl and HBr would be of much importance to the photochemistry studies as well as air quality improvement in urban areas of China.

**Author Contributions**

LY and YJT designed the research. XF, LY, YJT, JC, CY, YG, CL, KRD, FZ, ZL, BC, YM, LD, WD, JK, JTK, JZ, QZ, TK, SI, TP, DRW, VMK, YL, FB and MK carried out the observation, analyzed the data and interpreted the results. SI and TK provided quantum calculation results. XF, LY, YJT, and JC prepared the manuscript with contributions from all co-authors.

**Declaration of competing interest**

The authors declare that they have no known competing financial interests.

**Acknowledgement**

The work is supported by Academy of Finland (Center of Excellence in Atmospheric Sciences, project no. 307331, and PROFI3 funding, 311932), the European Research Council via ATM-GTP (742206) and the EMME-CARE project which has received funding from the European Union's Horizon 2020 Research and Innovation.



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





**Table 1.** Gas-phase acidities and deprotonated anion of a few compounds of interest.

| Compounds | Formula | $-\Delta G^a$ (kJ mol$^{-1}$) | Deprotonated Anion |
|---|---|---|---|
| Hydrobromic acid | HBr | 1319 | Br$^-$ |
| Nitric acid | HNO$_3$ | 1329 | NO$_3^-$ |
| hydrochloric acid | HCl | 1354 | Cl$^-$ |
| Nitrous Acid | HONO | 1396 | NO$_2^-$ |
| Isocyanic Acid | HCNO | 1415 | CNO$^-$ |
| Hydrocyanic Acid | HCN | 1433 | CN$^-$ |
| Hydroperoxy radical | HO$_2$ | 1450 | O$_2^-$ |
| Hypobromous Acid | HOBr | 1460 | BrO$^-$ |
| Hypochlorous Acid | HOCl | 1461 | ClO$^-$ |

*$^a$ Gas-phase acidity is defined as $-\Delta G$ for the protonation reaction ($H^+ + A^- \rightarrow HA$). Data are obtained from NIST Chemistry WebBook.*








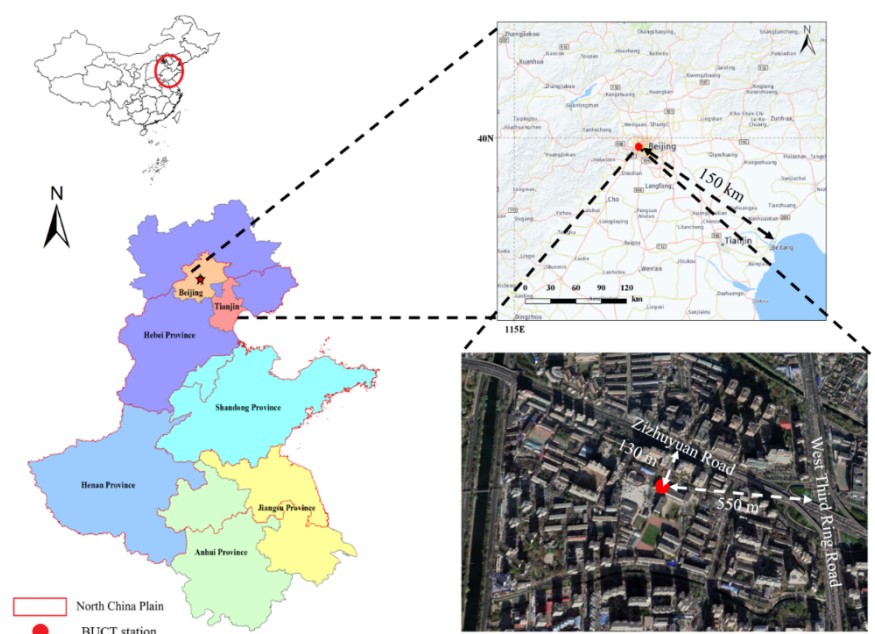

**Figure 1. The location of BUCT measurement station. The satellite map was revised from © Yahoo map and © Google map.**










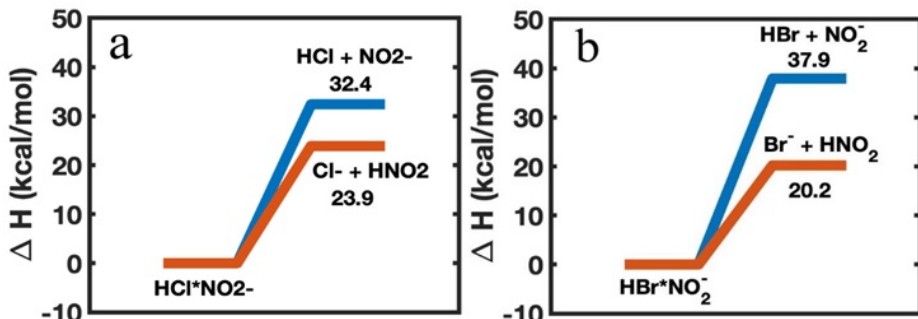

**Figure 2. The enthalpy of HCl·NO₂⁻ formed by HCl with NO₂⁻ and Cl⁻ with HNO₂ (a) and the enthalpy of HBr·NO₂⁻ formed by HBr with NO₂⁻ and Br⁻ with HNO₂ (b) calculated at the DLPNO-CCSD(T)/def2-QZVPP// ωB97X-D/aug-cc-pVTZ-PP level of theory.**







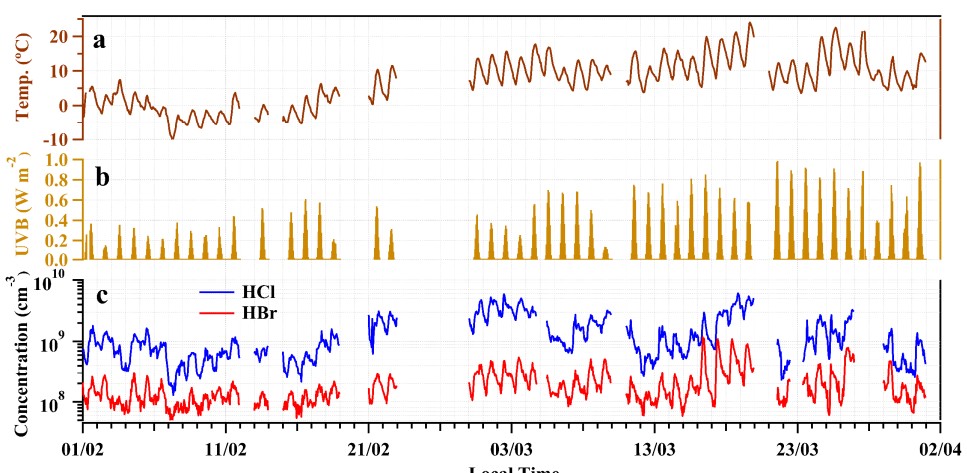


**Figure 3. Time series of temperature (a), UVB (b) and concentrations of hydrochloric acid (HCl) and hydrobromic acid (HBr) measured by the CI-APi-LTOF (c). The data points are in hourly-average interval.**




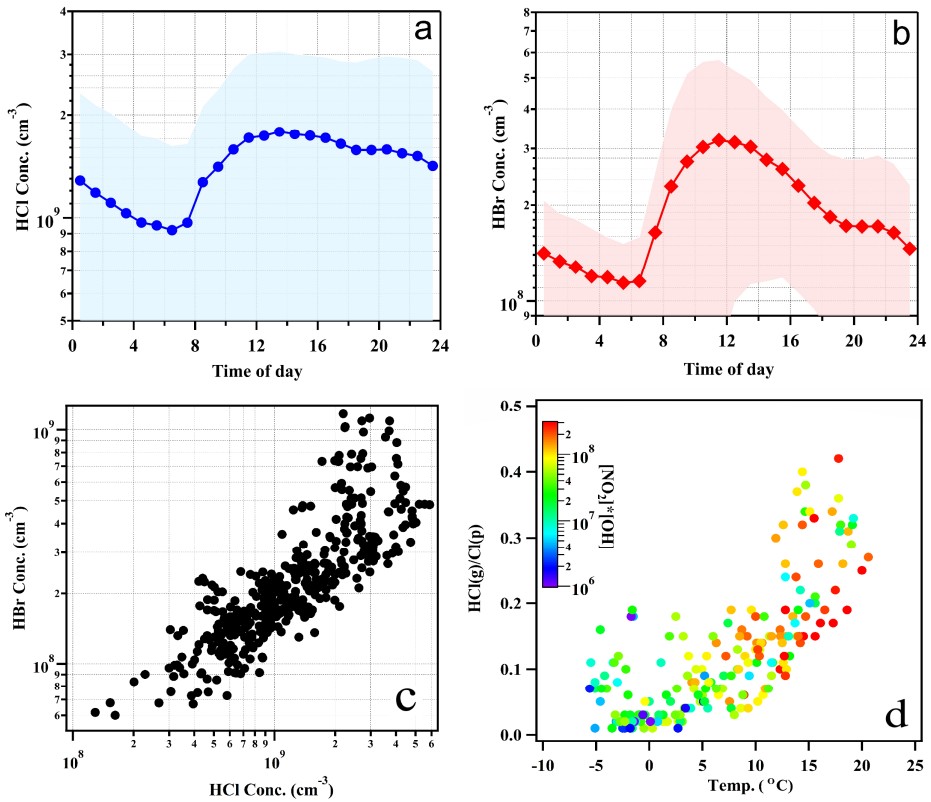

**Figure 4. Diurnal variations of HCl and HBr concentrations (averaged values ± one standard deviation) (a and b) and the correlation between HCl and HBr (c). In panel c, the data points are daytime (8:00-17:00) hourly averaged ones. All snowy and rainy days were excluded. Temperature dependence of heterogeneous reaction in HCl, coloured by the abundance of $HNO_3$ which was indicated by $[NO_2]*[OH]$ (d).**





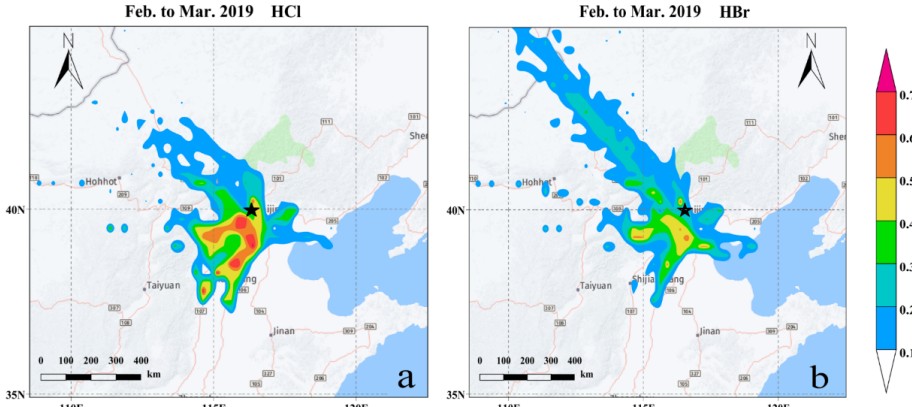

**Figure 5. The 24h back trajectories of HCl (a) and HBr (b) concentration higher than 75% quantile at 100 m height in February and March using MeteoInfo PSCF modelling (Wang, 2014, 2019). The colour bar shows the weight among all backward trajectories arriving at BUCT, Beijing (marked as a black star).**







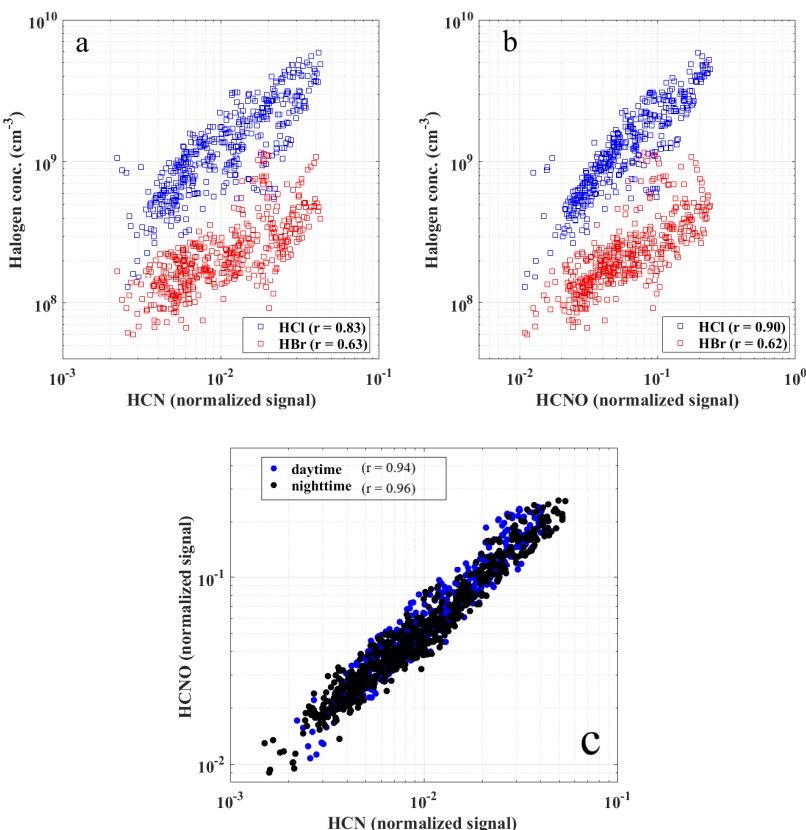

**Figure 6. The relationship of HCl and HBr concentrations with HCN (a) and HCNO (b) during the daytime and the correlations between HCN and HCNO during both daytime (08:00-17:00) and nighttime (18:00-07:00 the next day) (c).**






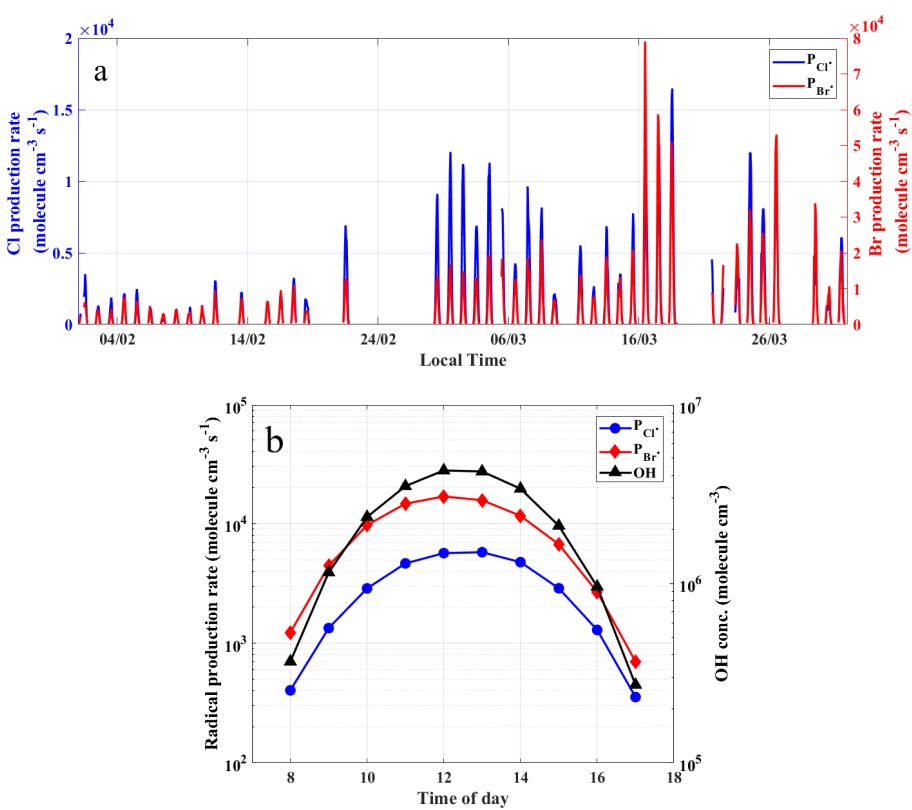


**Figure 7. Time series (a) and diurnal pattern (b) of HCl, HBr radical production rate and OH concentration during the observation period.**
