# Peer review of "Atmospheric gaseous hydrochloric and hydrobromic acid in urban Beijing, China: detection, source identification and potential atmospheric impacts"

_Atmospheric Chemistry and Physics, 2020_

## Referee Comment (RC1) · Anonymous Referee #1 · 7 Jan 2021

Gaseous hydrochloric and hydrobromic acid play important roles in tropospheric physicochemical processes, however, the atmospheric gaseous HCl and HBr in urban environments, are much less studied. This manuscript focused on the concurrent measurement of gaseous HCl and HBr by a CI-APi-LTOF mass spectrometer in urban Beijing, China, which is rarely reported before. Strong gaseous HCl and HBr were observed in Beijing where marine sources only have limited influence. Anthropogenic emissions seem to be a more important factor. This study estimates the production of atomic Cl and Br by the reactions of HCl and HBr with OH, which further contribute to atmospheric oxidation capacity. It provides a new insight of halogen chemistry in Chinese megacities and fits the scope and the interest of the journal of ACP. It is well organized and professionally written. Therefore, I recommend this manuscript for publication after minor revisions.

Some minor comments:

1. Line 5: In the authors list, It seems to be press error that some affiliations are marked with superscript and the rest ones are with subscript.

2. Line 115: Besides $HNO_3$, some other strong acids such as gaseous $H_2SO_4$ can also displace HX from sea-salt particles (Thornton et al., 2010). The authors should also add that information.

3. Line 174 delete the words of "atmospheric Br".

4. Line 178 "Crisp et al. (Crisp et al., 2014) summarized that…" should be revised to "Crisp et al. summarized that…"

5. To better present the results, I recommend the authors to improve the quality of the figures. Take Figure 6 for example, to better present the comparison, panel C can be divided into two sub-panels. The sizes of the panels of Figure 4, Figure S9 and S10 should be the same.

6. In the manuscript, to compare the concentrations of HCl and HBr between the Beijing, China in this study and other locations from previous ones (Lee et al., 2018;Simpson et al., 2015), it is better to also include the unit of mixing ratios (i.e., ppt) beside number concentrations (# $cm^{-3}$) in the measurement.

7. How do HCl and HBr behave on clean days and polluted days? are their concentrations higher during polluted days?

8. The title of "3.3 halogens atom production" should be "3.3 halogens' atom productions" or "3.3 halogen-atom productions". Besides, this part is interesting and important. Can

the authors expend this part a bit to better elucidate the potential applications of the results from the measurement?

9. In SI, a good correlation was observed between measured $J_{NO2}$ and modelled $J_{NO2}$ (Figure S11). Please add the brief description of the model that used.

References:

Lee, B. H., Lopez-Hilfiker, F. D., Schroder, J. C., Campuzano-Jost, P., Jimenez, J. L., McDuffie, E. E., Fibiger, D. L., Veres, P. R., Brown, S. S., Campos, T. L., Weinheimer, A. J., Flocke, F. F., Norris, G., O'Mara, K., Green, J. R., Fiddler, M. N., Bililign, S., Shah, V., Jaegle, L., and Thornton, J. A.: Airborne Observations of Reactive Inorganic Chlorine and Bromine Species in the Exhaust of Coal-Fired Power Plants, J Geophys Res Atmos, 123, 11225-11237, 10.1029/2018JD029284, 2018.

Simpson, W. R., Brown, S. S., Saiz-Lopez, A., Thornton, J. A., and Glasow, R.: Tropospheric halogen chemistry: sources, cycling, and impacts, Chem Rev, 115, 4035-4062, 10.1021/cr5006638, 2015.

Thornton, J. A., Kercher, J. P., Riedel, T. P., Wagner, N. L., Cozic, J., Holloway, J. S., Dube, W. P., Wolfe, G. M., Quinn, P. K., Middlebrook, A. M., Alexander, B., and Brown, S. S.: A large atomic chlorine source inferred from mid-continental reactive nitrogen chemistry, Nature, 464, 271-274, 2010.

---

## Referee Comment (RC2) · Anonymous Referee #2 · 23 Jan 2021

Fan et al present measurements of HCl and HBr in Beijing during winter. This is a unique dataset for which publication is worthwhile. My comments below mainly focus on clarifications and reporting uncertainties.

Measurement uncertainties need to be reported and shown throughout (including in main text and in figures), with the proper significant figures reported reflecting this uncertainty. In particular, often HBr values are presented quantitatively and with more certainty (over-reported in terms of significant figures) than is appropriate when the methods (Line 259) state that HBr concentrations should be treated as semi-quantitative (fix

word in text); for example, HBr is reported with 2 significant figures even at the single ppt level on Line 52 in the abstract, giving the impression of much greater certainty in this number. When error is reported, it should be with 1 significant figure (unlike on Line 286, where it should also be clarified that this is the standard deviation, I believe?). In addition, for improved comparisons to other work and other atmospheric species, it would be helpful for HCl and Br to be reported as mole ratios, in addition to the units of molec cm-3 (also please add "molec" before cm-3 throughout for clarity); this can be done in parentheses following the current units of molec cm-3, for example. Also, I did not see background/blank measurements discussed, and measurement limits of detection need to be reported.

Please clarify the discussion on Lines 287-291 about the correlations between HCl and HBr with temperature and UVB, as the discussion is limited and it is difficult to discern these points from the figures provided. Also, the authors state that "HCl and HBr concentrations being to increase together with the rising of temperature and UVB during April 2019", but the corresponding Figure 3 only shows data until April 2. Further, in the data shown, the concentration appears to decrease in late March even with higher temperature, showing the opposite of what is discussed in the text.

Throughout the manuscript (including the SI), please clarify whether the particulate chloride discussed is from the ACSM (non-refractory PM1 chloride) or MARGA (total PM2.5 chloride). This differentiation in which particulate chloride is shown may impact the interpretation of the results. It would seem most appropriate to use the MARGA data, but the methods section seems to imply that ACSM data is presented, although this is not clear. Regardless of which data are presented, justification and a quantitative comparison of the ACSM and MARGA chloride data need to be included in the SI. Also, please clarify the text on Lines 276-277, as it suggests as written that the ACSM measured PM2.5 chloride, which is not the case for the standard instrument.

Figure S9 shows the temporal variations in HCl compared to particulate chloride, and this is a key contribution to this work. Therefore, I highly suggest moving this figure to

the main text, with the following suggested modifications. Please state in the caption whether ACSM or MARGA Cl- is presented in the figure. In part b, plot both HCl and Cl- as mole ratios for improved comparison and evaluation of the assertion of gas-particle portioning dominating the diel profile. Then use these mole-based values for a more quantitative discussion on page 7.

It is stated on Lines 294-295 that HCl and HBr began to increase after sunrise, but the diel plots of Figure 4 do not include radiation for evaluation of this statement. Further, while Figure 3 shows these data as a stacked plot, it is too zoomed out to be able to allow evaluation of this statement. I suggest adding radiation to Figure 4.

HBr, similar to HCl, is also formed from the reaction of bromine atoms with hydro-carbons, which would be expected to occur during daylight, when temperature is also higher. It is not clear how the authors are differentiating this process from gas-particle partitioning. Please clarify.

The 24 h air mass trajectory analysis for HBr is not consistent with the typical atmo-spheric lifetime of 2.5 h reported on Line 129. What is the estimated atmospheric lifetime of HBr under the conditions of this study? Please make sure that the length of the air mass trajectory analysis is appropriate considering the lifetime.

Additional Comments: Line 56 & elsewhere: Please change "gas-aerosol partitioning" to "gas-particle partitioning".

Introduction, Paragraph 1: The authors are encouraged to also include mercury in the motivation to study atmospheric bromine chemistry.

Lines 142-144: Additional inland measurements of chlorine chemistry include Mielke et al. 2016 (Can. J. Chem.) and McNamara et al. 2020 (ACS Central Sci.).

Lines 164-165: Please clarify this sentence as it not clear what species are included here, and the manuscript title mentions the stratosphere.

Lines 178-181 and 312: Note that McNamara et al. 2020 (ACS Central Sci.) also

reported inland HCl measurements.

Line 223: Do the authors mean 5 s averaging here, as TOFs operate with much higher time resolution than 5 s.

Line 302: Please add explanation of Figure 4d to this paragraph to improve the clarity of the discussion here.

Figure 4: Please consider showing Parts a and b on linear scales in ppt. Part d needs more explanation in the caption. Is this a mass or mole ratio? Also, I did not see measurements of NO2 in the methods section, so are both NO2 and OH calculated values here? Please clarify here in the caption, on Line 374, and SI Line 275 how NO2 was obtained.

Lines 364-366: Please clarify where these measurements were made, as this context is needed for understanding the relevance of the statement.

Lines 366-368: Please clarify these sentences. What is meant by extractable gaseous organic bromine? What is the statement "high concentration and reactivity of both organic/inorganic Br" referring to here (as only HBr is presented in this work and organic Br is generally not very reactive)?

Lines 375-378: Note that McNamara et al 2020 (ACS Central Sci) also reported similar Cl atom production rates from OH + HCl for the inland urban environment.

Lines 382-383: Please provide references for this statement of previous work and also compare the Cl atom production rates from HCl in this work to ClNO2 and Cl2 photolysis in the previous studies.

Line 387: This statement about the ubiquity of bromine chemistry in the polluted urban environment is speculative and not supported by the paper cited.

Lines 499-503: This reference is listed twice.

Figure 1: The font on these maps is not readable as currently presented.
* * *
Interactive
comment

Figure 2: Consider showing HCl on the left axis and HBr on the right axis, both on linear scales. Incorporate measurement uncertainty.

Figure 7: Please clarify in the caption that all of the 'data' shown are calculations, not measurements, as implied by the text in the caption.

---

## Author Response (AR1)

**Reviewer(s)' Comments to Author:**

**Reviewer: 1**

*Comments:*

*General Comment*

*Gaseous hydrochloric and hydrobromic acid play important roles in tropospheric physicochemical processes, however, the atmospheric gaseous HCl and HBr in urban environments, are much less studied. This manuscript focused on the concurrent measurement of gaseous HCl and HBr by a CI-APi-LTOF mass spectrometer in urban Beijing, China, which is rarely reported before. Strong gaseous HCl and HBr were observed in Beijing where marine sources only have limited influence. Anthropogenic emissions seem to be a more important factor. This study estimates the production of atomic Cl and Br by the reactions of HCl and HBr with OH, which further contribute to atmospheric oxidation capacity. It provides a new insight of halogen chemistry in Chinese megacities and fits the scope and the interest of the journal of ACP. It is well organized and professionally written. Therefore, I recommend this manuscript for publication after minor revisions.*

Reply: We are very grateful to the positive comments and helpful suggestions from Reviewer #1, and have carefully revised our manuscript accordingly. A point-to-point response to reviewers' comments, which are repeated in italic, is given below.

*Comments:*

*1. Line 5: In the authors list, it seems to be press error that some affiliations are marked with superscript and the rest ones are with subscript.*

Reply: Thanks for pointing it out. We have corrected this typo in our revised manuscript.

*2. Line 115: Besides $HNO_3$, some other strong acids such as gaseous $H_2SO_4$ can also displace HX from sea-salt particles (Thornton et al., 2010). The authors should also add that information.*

Reply: We have added this information to our revised manuscript.

Revised text in the main text (line 112-115):

"It is known that sea-salt particle is a major source of atomic halogens in the marine environment. The chloride ($Cl^-$) and bromide ($Br^-$) in the sea-salt particles can be displaced by strong acids (i.e., nitric acid ($HNO_3$) and sulfuric acid ($H_2SO_4$)) to release gas-phase hydrogen halides HX (reaction (R1); X = Cl or Br) into the atmosphere (Gard et al., 1998;Thornton et al., 2010)."

*3. Line 174 delete the words of "atmospheric Br".*

Reply: We have removed the words "atmospheric Br".

Revised text in main text (line 173-176):

"Yet, since the phasing out of leaded gasoline, the long-term atmospheric Br exhibited a continuous decreasing trend for 2 to 3 decades in Germany (Lammel et al., 2002), and a similar situation is expected in Beijing as the usage of leaded gasoline was banned from the years around the 2000s in China (Cai et al., 2017)."

*4. Line 178 "Crisp et al. (Crisp et al., 2014) summarized that…" should be revise to "Crisp et al. summarized that…".*

Reply: We have revised accordingly.

Revised text in the main text (line 179-184):

"Some limited studies focused on the atmospheric HCl, for example, Crisp et al. (2014) summarized that the concentration of HCl is typically less than 1 ppb over the continental regions and McNamara et al. (2020) measured the concentration of HCl is around 100 ppt from inland sources, while an airborne measurement showed HCl concentrations of around 100 ppt was typically observed over the land area of northeast United States, except near power plant plumes with concentrations over 1 ppb (Crisp et al., 2014;McNamara et al. 2020;Haskins et al., 2018)."

*5. To better present the results, I recommend the authors to improve the quality of the figures. Take Figure 6 for example, to better present the comparison, panel C can be divided into two sub-panels. The sizes of the panels of Figure 4, Figure S9 and S10 should be the same.*

Reply: Thank you for your advice. These figures have already been improved accordingly. Figure S9 has been moved to the revised main text as Figure 5. The original Figure 6 and Figure S10 were changed to Figure 7 in the revised main text and Figure S9 in revised SI, respectively. To make it easier to view, these revised figures were attached below.

[Figure]

**Figure R1 (Figure 4 in the revised main text).** Diurnal variations of UVB intensities, HCl and HBr concentrations (averaged values ± one standard deviation) (**a** and **b**) and the correlation between HCl and HBr (**c**). In panel c, the data points are hourly averaged ones during daytime (8:00-17:00). Temperature dependence of gas to particle partitioning ratios of mass concentration of chloride, colour-coded by [NO$_2$]*[OH] which was indicated as the abundance of HNO$_3$ (**d**). All snowy and rainy days during the sampling period were excluded.

[Figure]

**Figure R2 (Figure 5 in the revised main text).** Time profile of daily averaged concentration of particulate chloride (Cl(p)) measured by ACSM, gaseous HCl (HCl(g)) measured by CI-APi-LTOF and the mole ratio of HCl(g)/Cl(p) (**a**) and diurnal variation of HCl(g), Cl (p) and HCl(g)/Cl(p) (**b**).

[Figure]

**Figure R3 (Figure 7 in the revised main text).** The relationship of HCl and HBr concentrations with HCN (**a**) and HCNO (**b**) during the daytime (08:00-17:00) and the correlations between HCN and HCNO during both daytime (08:00-17:00) (**c**) and nighttime (18:00-07:00 the next day) (**d**). The data points are hourly averaged ones.

[Figure]

**Figure R4 (Figure S10 in the revised SI).** The correlation ($r = 0.67$) between hourly mean mass concentrations of particulate Cl (Cl(p)) and black carbon (BC) (**a**); correlations between daily mean concentrations of HCl ($r = 0.82$), HBr ($r = 0.60$) and BC during observation periods from 1 February to 31 March, 2019 (**b**).

*6. In the manuscript, to compare the concentrations of HCl and HBr between Beijing, China in this study and other locations from previous ones (Lee et al., 2018; Simpson et al., 2015), it is better to also include the unit of mixing ratios (i.e., ppt) beside number concentrations (# cm$^{-3}$) in the measurement.*

Reply: We have included the mixing ratios as ppt to the text.

Revised text in the main text (line 52-54):

"We observed significant HCl and HBr concentrations ranged from a minimum value at $1 \times 10^8$

molecules cm$^{-3}$ (4 ppt) and $4\times10^7$ molecules cm$^{-3}$ (1 ppt) up to $6\times10^9$ molecules cm$^{-3}$ (222 ppt) and $1\times10^9$ molecules cm$^{-3}$ (37 ppt), respectively."

Revised text in the main text (line 302-304):

"The mean concentrations of HCl and HBr are $1\times10^9$ molecules cm$^{-3}$ (37 ppt) and $2\times10^8$ molecules cm$^{-3}$ (7 ppt), respectively. The maximum concentrations reach up to $6\times10^9$ molecules cm$^{-3}$ (222 ppt) for HCl, and $1\times10^9$ molecules cm$^{-3}$ (37 ppt) for HBr during the daytime."

*7. How do HCl and HBr behave on clean days and polluted days? are their concentrations higher during polluted days?*

Reply: HCl and HBr have different behaviors between clean days and polluted days during our observation period. HCl and HBr are more abundant on haze days (daily mean PM$_{2.5} \geq$ 75 ug/m$^3$) than that on clean days (daily mean PM$_{2.5} <$ 75 ug/m$^3$). The detail has been added in "3.3 halogen-atom productions." Please also refer to the reply below for Comment 8.

*8. The title of "3.3 halogens atom production" should be "3.3 halogens' atom productions" or "3.3 halogen-atom productions". Besides, this part is interesting and important. Can the authors expend this part a bit to better elucidate the potential applications of the results from the measurement?*

Reply: The title "3.3 halogens atom production." has been changed to "3.3 halogen-atom productions.".

Also, this section has been expanded accordingly (Line 414-419):

"The average HCl and HBr concentrations were observed to be higher during the polluted days (daily mean PM$_{2.5} \geq$ 75 μg/m$^3$), which is about 2-3 times higher than the clean days (daily mean PM$_{2.5} <$ 75 μg/m$^3$), as shown in Figure 8b. Consequently, the radical production rate also showed a difference between clean and polluted days (Figure 8d). The daily mean value of $P_{Cl}$. (up to $8\times10^3$ molecules cm$^{-3}$ s$^{-1}$) and $P_{Br}$. ($2\times10^4$ molecules cm$^{-3}$ s$^{-1}$) in polluted days were both higher than that of clean days by up to 2 times. This hints that the roles of HCl and HBr may be more significant in polluted environments."

[Figure]

**Figure R5 (Figure 8 in the revised main text).** Time series of calculated production rates of Cl and Br radicals during the observation period (**a**); diurnal variations of HCl and HBr concentrations in clean and polluted days (**b**); diurnal variations of production rates of Cl and Br radicals, together with calculated OH radical concentrations (**c**) and production rates of Cl and Br radicals in clean and polluted days (**d**). The clean and polluted days were classified as daily $PM_{2.5} < 75$ µg/m$^3$ and $PM_{2.5} \geq 75$ µg/m$^3$, respectively. The data points are in the hourly-average interval and measured during observation periods from 1 February to 31 March 2019.

*9. In SI, a good correlation was observed between measured $J_{NO2}$ and modelled $J_{NO2}$ (Figure S11). Please add the brief description of the model that used.*

Reply: In this study, unfortunately, direct measurement of $J_{NO2}$ is not available during our observation periods. The photolysis rate constants of $NO_2$ ($J_{NO2}$) were calculated according to the solar zenith angle and the location using a box model (FACSIMILE 4) (Liu et al., 2020). Then, the output (modelled $J_{NO2}$) was compared to the measured $J_{NO2}$ (by using $NO_2$ photolysis sensor ($J_{NO2}$, Metcon)) for the period when $J_{NO2}$ measurements were available, as shown in Figure S10, to ensure that the modelled $J_{NO2}$ is relevant. We have added the brief description of the model into the SI Section S8 "The calculations of OH concentration and production rate of atomic Cl and Br".

Revised text in SI (line 365-368):

"Photolysis rate constants of $NO_2$ ($J_{NO2}$) were calculated according to the solar zenith angle and the location using a box model (FACSIMILE 4) (Liu et al., 2020). Using another dataset collected from 21 May to 10 June 2019, a good correlation ($r$=0.97) between calculated and measured $J_{NO2}$ confirmed the validation of our calculated $J_{NO2}$ (Figure S11a).

**References:**

Liu, Y., Zhang, Y., Lian, C., Yan, C., Feng, Z., Zheng, F., Fan, X., Chen, Y., Wang, W., Chu, B., Wang, Y., Cai, J., Du, W., Daellenbach, K. R., Kangasluoma, J., Bianchi, F., Kujansuu, J., Petäjä, T., Wang, X., Hu, B., Wang, Y., Ge, M., He, H., and Kulmala, M.: The promotion effect of nitrous acid on aerosol formation in wintertime in Beijing: the possible contribution of traffic-related emissions, Atmos. Chem. Phys., 20, 13023-13040, 10.5194/acp-20-13023-2020, 2020.

**Reviewer(s)' Comments to Author:**

**Reviewer: 2**

*Comments:*

*General Comment*

*Fan et al present measurements of HCl and HBr in Beijing during winter. This is a unique dataset for which publication is worthwhile. My comments below mainly focus on clarifications and reporting uncertainties.*

Reply: We are very grateful for the positive comments and helpful suggestions. And we have carefully revised our manuscript accordingly. A point-to-point response to reviewers' comments (in italic), is given below.

*Major Comments:*

*1. Measurement uncertainties need to be reported and shown throughout (including in main text and in figures), with the proper significant figures reported reflecting this uncertainty. In particular, often HBr values are presented quantitatively and with more certainty (over-reported in terms of significant figures) than is appropriate when the methods (Line 259) state that HBr concentrations should be treated as semi-quantitative (fix word in text); for example, HBr is reported with 2 significant figures even at the single ppt level on Line 52 in the abstract, giving the impression of much greater certainty in this number. When error is reported, it should be with 1 significant figure (unlike on Line 286, where it should also be clarified that this is the standard deviation, I believe?).*

Reply: Thanks for the suggestion. We agree with the reviewer that reporting the uncertainty and showing proper significant figures are important. We added the information of the measurement uncertainty calculations as suggested in SI (Section S5. Calibration factor and Uncertainty estimation) line 261-282:

The uncertainties of HCl and HBr measurement come from two parts: (1) the HCl measurement uncertainties coming from MARGA ($\delta_{MARGA}$) and (2) fitting errors coming from the intercomparison between MARGA and CI-APi-LTOF measurements ($\delta_{MARGA-LTOF}$). The total uncertainties were calculated with the uncertainties propagation equation shown in Eq. (1). The $\delta_{MARGA}$ was obtained from many previous studies and assumed to be 15% (Trebs et al., 2004; Du et al., 2010, 2011; Wang et al., 2013). The $\delta_{MARGA-LTOF}$ was calculated through a procedure implemented in MATLAB (Mathworks, Inc.) from the fitting errors between HCl concentrations measured by MARGA and normalized HCl signals from CI-APi-LTOF in the intercomparison from the period of 2 to 6 September 2019 (shown in Table R1 and Figure S8). $\delta_{MARGA-LTOF}$ varied from 11% to 24% according to the fitting calculations (with 95% confidence bounds). To be more conservative, $\delta_{MARGA-LTOF}$ of 0.24 was applied into Eq. (1) to calculate the

total uncertainty ($\delta$). Therefore, the total uncertainty of $\pm 30\%$ was estimated in reporting HCl concentrations. The uncertainties for HBr were assumed to be the same with HCl since it is assumed that HCl and HBr have the same sensitivities for CI-APi-LTOF.

$$\delta = \sqrt{\delta_{MARGA}^2 + \delta_{MARGA-LTOF}^2} = \sqrt{(0.15)^2 + (0.24)^2} = 0.28 \qquad (1)$$

It should be noted that our assumptions of uncertainties could be regarded as a lower limit of HCl and HBr measurement since other potential uncertainty factors (e.g., different sensitivities of HCl and HBr) were not taken into account.

**Table R1 (Table S2 in revised SI). Estimated uncertainty factors in HCl quantification.**

|  | Relative uncertainty ($\delta$) |
|---|---|
| MARGA measurement | 0.15 |
| Intercomparison between MARGA and CI-APi-TOF | 0.24 |

Meanwhile, we have added the uncertainty discussion to the main text and revised some statements accordingly to report proper significant figures.

Revised text in the main text:

Line 52-54:

"We observed significant HCl and HBr concentrations ranged from a minimum value at $1\times10^8$ molecules cm$^{-3}$ (4 ppt) and $4\times10^7$ molecules cm$^{-3}$ (1 ppt) up to $6\times10^9$ molecules cm$^{-3}$ (222 ppt) and $1\times10^9$ molecules cm$^{-3}$ (37 ppt), respectively."

Line 248-253:

"The obtained calibration factor $C_{HCl}$ for HCl is $3 \pm 0.1\times10^{12}$ molecules cm$^{-3}$ (Figure S8b) and the uncertainty of $\pm 30\%$ (Section S5) was applied to the reported HCl concentrations. Similar to HCl, the same uncertainty was also adopted for HBr mixing ratios. It should be noted that our assumptions lead towards a lower limit estimate of HCl and HBr concentrations, due to other potential uncertainties (e.g., different sensitivities of HCl and HBr) were not taken into account."

Line 302-304:

"The mean concentrations of HCl and HBr are $1\times10^9$ molecules cm$^{-3}$ (37 ppt) and $2\times10^8$ molecules cm$^{-3}$ (7 ppt), respectively. The maximum concentrations reach up to $6\times10^9$ molecules cm$^{-3}$ (222 ppt) for HCl, and $1\times10^9$ molecules cm$^{-3}$ (37 ppt) for HBr during the daytime."

*2. In addition, for improved comparisons to other work and other atmospheric species, it would be helpful for HCl and HBr to be reported as mole ratios, in addition to the units of molec cm$^{-3}$ (also please add "molec" before cm$^{-3}$ throughout for clarity); this can be done in parentheses*

*following the current units of molec cm$^{-3}$, for example. Also, I did not see background/blank measurements discussed, and measurement limits of detection need to be reported.*

Reply: The concentration unit was revised from "cm$^{-3}$" to "molecules cm$^{-3}$" throughout the whole manuscript.

In this study, background/blank measurement was conducted by zero air measurement and the limits of detection (LODs) were defined as 3 times the standard deviation of background signals within a 2 min integration time. Experimental details for background/blank measurement and detection limits estimation were introduced in Section S4 in SI (Section S4 Laboratory experiment of HCl and HBr).

From the background measurement, the normalized background signals of HCl and HBr were found to be $7\times10^{-5}$ and $1\times10^{-5}$, which represented $1\times10^{8}$ and $1\times10^{7}$ molecules cm$^{-3}$ (i.e., 4 and 0.5 ppt) for HCl and HBr, respectively. We have added the statements of background/blank measurements and LODs in the revised manuscript.

Revised text in Supporting Information (Section S4 Laboratory experiment of HCl and HBr)

Revised text in SI (line 212-219):

"In order to quantitively confirm HCl and HBr can be detected by CI-APi-LTOF, 1 ml liquid standard HCl (Gaosheng, 36%) and HBr (Macklin, 48%) were diluted in 1000 ml MILLIPORE® ultrapure water resulting in $1.2\times10^{-2}$ mol L$^{-1}$ HCl and $8.6\times10^{-3}$ mol L$^{-1}$ HBr, respectively. Then, 10 ml diluted HCl and HBr solution was put in a U-shape glass tube and then the mixed flow of evaporated HCl and HBr and zero air (1L min$^{-1}$) went into CI-APi-LTOF (Figure S6). After the injection of gaseous HCl and HBr, the signals of Cl$^{-}$, Br$^{-}$, Cl$^{-}\cdot$HNO$_3$ (or HCl$\cdot$NO$_3^{-}$) and Br$^{-}\cdot$HNO$_3$ (or HCl$\cdot$NO$_3^{-}$) started to increase (Figure S7), confirming that the HCl and HBr can be detected as Cl$^{-}$, Br$^{-}$, Cl$^{-}\cdot$HNO$_3$ (or HCl$\cdot$NO$_3^{-}$) and Br$^{-}\cdot$HNO$_3$ (or HBr$\cdot$NO$_3^{-}$) by CI-APi-LTOF."

Revised text in the main text (line 243-246):

"The background measurement was carried out by sampling zero air. From Figure S7, the background signals were significantly lower than that of ambient air and injected HCl and HBr. The limits of detection (LODs, 3σ) were $1\times10^{8}$ and $1\times10^{7}$ molecules cm$^{-3}$ (i.e., 4 and 0.5 ppt) for HCl and HBr, respectively."

[Figure]

**Figure R1 (Figure S6 in revised SI)**. Schematic of the laboratory experiment set up.

[Figure]

**Figure R2 (Figure S7 in revised SI).** Time profile of normalized $Cl^-$, $Br^-$, $Cl^- \cdot HNO_3$ (or $HCl \cdot NO_3^-$) and $Br^- \cdot HNO_3$ (or $HBr \cdot NO_3^-$) signals measured by CI-APi-LTOF during the laboratory test. Within the duration from 0 to 12min, ambient air was sampled and measured by CI-APi-LTOF. The signals of zero air generated from a pure air generator (AADCO 737) were treated as background signals. The detection limits of HCl and HBr were obtained three times the standard deviation of background signals within a 2 min integration time. After the injection of gaseous HCl and HBr which evaporated from HCl and HBr solution, the elevated $Cl^-$, $Br^-$, $Cl^- \cdot HNO_3$ (or $HCl \cdot NO_3^-$) and $Br^- \cdot HNO_3$ (or $HBr \cdot NO_3^-$) signals were observed.

*3. Please clarify the discussion on Lines 287-291 about the correlations between HCl and HBr with temperature and UVB, as the discussion is limited and it is difficult to discern these points*

*from the figures provided. Also, the authors state that "HCl and HBr concentrations being to increase together with the rising of temperature and UVB during April 2019", but the corresponding Figure 3 only shows data until April 2. Further, in the data shown, the concentration appears to decrease in late March even with higher temperature, showing the opposite of what is discussed in the text.*

Reply: To make the sampling period clear, we emphasized our sampling period was from 1 February to 2 April 2019 in the main text and Figure 3 in the revised manuscript.

As pointed out by the reviewer, we did notice that "the concentration appears to decrease in late March even with higher temperature". Yet, the absolute HCl and HBr concentrations are assumed to be affected not only by temperature and UVB, but also by the photochemistry and gas-particle partitioning, indicating by the abundance of $HNO_3$ (indicated as $[NO_2]*[OH]$) and particulate chloride (Cl (p)), respectively. From revised Figure R3 (Figure 3 in the main text), although the temperature in late March is slightly higher than the beginning of February, the abundant of $HNO_3$ and particulate chloride in late March is comparatively lower than the beginning of February. The finding of lower Cl(p) concentration in spring than winter is similar to the previous literature in Beijing (Hu et al., 2017) and likely due to fewer combustion emissions. Thus, although in general, the gas/particle ratio increased with temperature (discussed in Major Comments 5 &6), the absolute HCl and HBr concentrations remain at a relatively low level in late March even with higher temperature.

To avoid potential misleading for the readers, we revised the main text as follows (line 305-310):

"For the first period of measurement (from 1 to 15 February), HCl and HBr concentrations are lower when the atmospheric temperature is close to 0°C and the UVB intensities are relatively low. Yet, for the later period of March, the HCl and HBr concentrations begin to increase along with the rising of temperature and UV. In late March, even with higher temperature, due to the less abundant of $HNO_3$ and particulate chloride, the HCl and HBr concentrations remain at a relatively low level (Figure 3)."

[Figure]

**Figure R3 (Figure 3 in revised main text).** Time profiles of temperature (**a**), UVB intensities (**b**), $[NO_2]*[OH]$ (**c**), particulate chloride (Cl(p)) (**d**), and the mixing ratios of HCl and HBr (**e**). The data points are in hourly-average interval.

*4. Throughout the manuscript (including the SI), please clarify whether the particulate chloride discussed is from the ACSM (non-refractory PM1 chloride) or MARGA (total PM2.5 chloride). This differentiation in which particulate chloride is shown may impact the interpretation of the results. It would seem most appropriate to use the MARGA data, but the methods section seems to imply that ACSM data is presented, although this is not clear. Regardless of which data are presented, justification and a quantitative comparison of the ACSM and MARGA chloride data need to be included in the SI. Also, please clarify the text on Lines 276-277, as it suggests as written that the ACSM measured PM2.5 chloride, which is not the case for the standard instrument.*

Reply: In this study, we applied an ACSM equipped with a $PM_{2.5}$ aerodynamic lens to present the particulate chloride (Cl(p)) due to the absence of MARGA data during the sampling period. Previous studies have demonstrated that the chloride from ACSM (non-refractory Cl) and online ion chromatography strongly correlated (Canagaratna et al., 2007). From our present study, a high correlation was also observed for particulate Cl between MARGA and ACSM for the later comparison (the same calibration period for CI-API-LTOF and MARGA, $r$=0.98, intercept=0.067, added in Figure S8). Thus, considering their high correlations, the application of using ACSM to measure particulate Cl during our sampling period is acceptable.

Moreover, the application of $PM_{2.5}$ aerodynamic lens for ACSM, which substantially increased the transmission efficiency for the particles with diameters over 1 µm, has been developed and applied in the heavily polluted areas (Xu et al., 2018; Peck et al., 2016; Zheng et al., 2020). Stable performances were achieved for $PM_{2.5}$ lens ACSM when compared with other online chemical component instruments such as ion chromatography and Sunset OC/EC Analyzer (Zhang et al., 2017).

[Figure]

**Figure R4 (Figure S8 in revised SI).** Time series of normalized HCl signals measured by CI-APi-LTOF and HCl(g) measured by MARGA during the calibration period (i.e., 2 to 6 September 2019) (**a**); The correlation between HCl(g) and normalized HCl signals measured by CI-APi-LTOF (**b**); Time series of particulate Cl (Cl(p)) measured by MARGA and ACSM (**c**); The correlation between particulate Cl (Cl(p)) measured by MARGA and ACSM (**d**).

To clarify the particulate Cl measurement, we revised the description and added the quantitative comparison for ACSM and MARGA to SI (Figure S8, shown above) as suggested in line 244-250. "A time-of-flight aerosol chemical speciation monitor (ToF-ACSM, Aerodyne Research Inc., USA) equipped with a PM$_{2.5}$ aerodynamic lens was applied to measure the particulate non-refractory Cl. The detailed introduction for ToF-ACSM with PM$_{2.5}$ lens and its comparison with other instruments could be found in previous literature (Xu et al., 2017;Zheng et al., 2020;Zhang et al., 2017). Although ACSM only measures the non-refractory Cl, a high correlation was achieved for the comparison between ACSM and MARGA measurement (*r*=0.98, intercept=0.067, shown in Figure S8d), suggesting non-refractory Cl measured by ACSM could explain a large proportion of particulate Cl in our sampling period."

*5. Figure S9 shows the temporal variations in HCl compared to particulate chloride, and this is a key contribution to this work. Therefore, I highly suggest moving this figure to the main text, with the following suggested modifications. Please state in the caption whether ACSM or MARGA Cl-is presented in the figure. In part b, plot both HCl and Cl- as mole ratios for improved comparison*

*and evaluation of the assertion of gas-particle portioning dominating the diel profile. Then use these mole-based values for a more quantitative discussion on page 7.*

Reply: Thanks for the reviewer's suggestion. Figure S9 has been moved to the main text (Figure 5) with the revised figure captions in the revised version. It is now clearly stated the Cl(p) is measured by ACSM. The original panel (c) was also presented as the mole ratio of HCl(g)/Cl(p) as suggested in the new Figure 5b. To better exhibit the quantitative discussions related to gaseous HCl and particulate Cl diurnal variations. We revised the discussion in the main text from line 327 to line 338.

"Our observation of daily averaged mass concentrations of particulate chloride (Cl (p)) in $PM_{2.5}$ showed a similar trend with daily averaged mixing ratios of gaseous HCl (Figure 5a). The difference from the ratios of HCl(g) to Cl(p) in February and March is likely due to the higher temperature in March (Figure 3 and 5a). In contrast, the diurnal variations of HCl and particulate Cl showed the opposite trend at daytime from 08:00 to 17:00 (Figure 5b). The mole ratios of HCl(g) to Cl(p) ranged from <0.1 at nighttime and early morning to >0.3 in the afternoon (Figure 5b). The enhancement of HCl(g)/Cl(p) during the noontime is owing to the large increase of gaseous HCl. It also suggested that the higher temperature and stronger photochemical reactions during the daytime would strongly influence HCl releases from particulate chloride in Beijing, which will be further discussed in the following discussions. During the period between the late afternoon and midnight, the increase of Cl(p) and HCl(g) could be explained by the higher nighttime emissions of residential combustions such as wood and coal burnings in Beijing (Hu et al., 2017;Sun et al., 2016) and high abundance of gaseous $HNO_3$ are attributed to efficient nocturnal $N_2O_5$ chemistry (Tham et al., 2018)."

[Figure]

**Figure R5 (Figure 5 in revised main text).** Time variation of daily averaged concentration of particulate chloride (Cl(p)) measured by ACSM, gaseous HCl (HCl(g)) measured by CI-APi-LTOF and mole ratios of HCl(g)/Cl(p) (**a**) and diurnal variation of HCl(g), Cl (p) and mole ratios of HCl(g)/Cl(p) (**b**).

*6. It is stated on Lines 294-295 that HCl and HBr began to increase after sunrise, but the diel plots of Figure 4 do not include radiation for evaluation of this statement. Further, while Figure 3 shows these data as a stacked plot, it is too zoomed out to be able to allow evaluation of this statement. I suggest adding radiation to Figure 4.*

Reply: Owing to the total radiations were not recorded during our sampling periods, the UVB intensities were used as alternative ones. The diel plot of UVB (280 - 315 nm) intensities was added to Figure 4 (Figure R6 in this reply) in the main text. From Figure 4 a and b, accompanied by the increasing UVB intensities, the mixing ratios of HCl and HBr started to elevate. The possible driving forces for this would be the increase of atmospheric temperature together with the elevated abundance of $HNO_3$. From Figure 4d, it is distinct that the partitioning ratios (HCl(g)/Cl(p)) of HCl were enhanced by the high temperature and high abundance of $HNO_3$, which was indicated as $[NO_2]*[OH]$. Due to the mass concentrations of particulate Br were not measured during our sampling period, a similar plot (Figure 4d) for HBr was not available. Nonetheless, based on the good correlation between HCl and HBr (Figure 4c) and similar diel variations, it is rational to propose that the HBr was also likely affected by the same factors, similar to HCl.

It also should be noted that, during the period between the late afternoon and midnight, the relatively high concentrations of gaseous HCl and HBr still can be observed. One possible reason for this could be that a relatively high abundance of $HNO_3$ formed from nocturnal $NO_3$ radical chemistry during this period. Further research is still needed to investigate the effect of $NO_3$ radical chemistry in the nighttime.

[Figure]

**Figure R6 (Figure 4 in the revised main text).** Diurnal variations of UVB intensities, HCl and HBr concentrations (averaged values ± one standard deviation) (**a** and **b**) and the correlation between HCl and HBr (**c**). In panel c, the data points are hourly averaged ones during daytime (8:00-17:00). Temperature dependence of gas to particle partitioning ratios of mass concentration of chloride, colour-coded by

[NO$_2$]*[OH] which was indicated as the abundance of HNO$_3$ (**d**). All snowy and rainy days during the sampling period were excluded.

*7. HBr, similar to HCl, is also formed from the reaction of bromine atoms with hydro-carbons, which would be expected to occur during daylight, when temperature is also higher. It is not clear how the authors are differentiating this process from gas-particle partitioning. Please clarify.*

Reply: Yes, the reviewer has a point here. Besides the contribution from gas-particle partitioning, the reaction of bromine atoms with hydrocarbons during the daytime could be another pathway for HBr formation. Unfortunately, without the observation data of other bromine species (e.g., Br$_2$, BrNO$_2$), it is impossible to differentiate and quantify the contribution by the reaction of bromine atoms with hydrocarbons. Although we cannot exclude the contribution by the reaction of bromine atoms with hydrocarbons to form HBr, this process is likely not the dominant pathway because:

1) Bromine atom is less reactive to hydrocarbons compared to the chlorine atom, and most often reacts with ozone, or reacts with aldehydes (e.g., formaldehyde, which is less abundant, about 20% of the total VOCs, in Beijing) (Simpson et al., 2015; Li et al., 2010).

2) The HBr shows a good correlation with temperature and the HNO$_3$ precursors ([NO$_2$]*[OH]), as shown in Figure R3, indicating that the gas-particle partitioning process may be an important process.

Therefore, to make the statement clearer, we decide to revise the statement and add the above information to the revised main text as following:

Line 314-326:

"From Figure 4d, it also can be found that elevated temperature and high abundance of HNO$_3$ which was indicated as [NO$_2$]*[OH] could intensify the HCl releases from particulate chloride in the daytime from 08:00 to 17:00. The OH radical concentrations were calculated using J$_{NO2}$ and J$_{O1D}$ (Section S8). This phenomenon is consistent with our observation results above where the increase of temperature and UVB could reinforce the formation of chemicals (e.g., HNO$_3$) that promote the gas-particle partitioning or directly increase gas-phase formation rate of HCl and HBr (Crisp et al., 2014;Riedel et al., 2012), thus further enhancing the HCl and HBr (Figure 3). Although there is no direct measurement of particulate bromide (Br), considering the similarity in diurnal patterns and good correlation ($r = 0.70$) between HBr and HCl (Figure 4c), and HBr tracking well with the temperature and [NO$_2$]*[OH] (see Figure 3), it is rational to suppose HBr also predominantly derived from gas-particle partitioning process. The contribution by the reaction of bromine atoms with hydrocarbons to form HBr is likely not the dominant pathway as bromine atom is less reactive to hydrocarbons compared to the chlorine atom, and most often reacts with ozone (Simpson et al., 2015)."

*8. The 24 h air mass trajectory analysis for HBr is not consistent with the typical atmospheric lifetime of 2.5 h reported on Line 129. What is the estimated atmospheric lifetime of HBr under the conditions of this study? Please make sure that the length of the air mass trajectory analysis is appropriate considering the lifetime.*

Reply: We noted that the lifetime of HBr was estimated as 2.5h in the previous statement, which is

indeed shorter than the 24h back-trajectory coupled Potential Source Contribution Function (PSCF) analysis. The original purpose of applying back trajectory and PSCF in this study is to point the potential source regions for the air masses that led to high-level concentrations of HBr and HCl during the sampling period. And added the following description and Figure R7 (Figure S9 in revised SI) as a new section (Section S6. PSCF Analysis) in SI line 303-320:

"In PSCF method applied in this study, only the high concentrations of HBr and HCl (>75[th] percentile, mainly around the middle of the day due to the diurnal patterns) were linked to the trajectories and presented. The 24-h trajectories analysis were conducted by HYSPLIT GDAS data with the ending height of 100 m. PSCF analysis (MeteoInfo PSCF modelling) links the air mass trajectories and high concentrations. The higher potentials of trajectory pathways led to high concentration indicates the possible source regions. Details of PSCF could be found in Wang et al., (2014&2019). The time-series of HBr concentration during the whole sampling period and those applied in the PSCF analysis were exhibited in Figure S9b. It is shown that only the extremely high HBr concentrations were included in the trajectory and PSCF analysis, coincidence with the period of heavy pollution indicated by the high $PM_{2.5}$ concentration level (>80 µg m$^{-3}$ and ~100 µg m$^{-3}$ for HBr and HCl, respectively). Meanwhile, the lower HBr concentrations during the nighttime and clean periods were not included in the PSCF analysis.

Additionally, Cl/Br-containing particles would have a longer lifetime in the atmosphere (from hours to weeks for fine particles in the troposphere (Seinfeld, 2003) and continuously influence the gaseous HBr concentration through gas-particle partitioning. Therefore, in this study, the 24h-backwards air mass trajectory and PSCF analyses were adopted to indicate the source regions of the polluted air masses that result in the high Cl and Br levels rather than the real-time origins of gaseous HCl/HBr in the atmosphere. A shorter time in trajectory analysis, on the other hand, would increase the uncertainties in calculations and may not provide more information on source regions of air parcels at a large scale."

[Figure]

**Figure R7 (Figure S9 in revised SI).** Time series of gaseous HCl during the whole sampling period, those in PSCF analysis (>75th percentile) and PM2.5 mass concentrations (**a**), and gaseous HBr during the whole sampling period, those in PSCF analysis (>75th percentile) and PM2.5 mass concentrations (**b**).

To better explain the back trajectory and PSCF analysis, we added the following description in Line 290-296:

"we applied 24h air mass back trajectory and Potential Source Contribution Function (PSCF) analyses to help to elucidate the potential source regions (i.e., air masses) of high levels of HCl and HBr. The detailed descriptions of PSCF and air mass trajectory analysis were described in the SI (Section S6) and previous literature (Wang et al., (2014, 2019b)). It is noted that the lifetime of gaseous HCl and HBr could be shorter than the length of the air mass trajectories. These analyses mainly aimed to point out the source regions of pollutant air masses that brought high levels of Cl and Br rather than the real-time origins of air parcels."

We also revised the figure captions of Figure R8 (Figure 6 in the revised version) accordingly.

[Figure]

**Figure R8 (Figure 6 in revised main text).** The results of PSCF analysis for HCl (**a**) and HBr (**b**). Black stars mark the location of the sampling site.

**Minor comments**

*1. Line 56 & elsewhere: Please change "gas-aerosol partitioning" to "gas-particle partitioning".*
Reply: "gas-aerosol partitioning" has been changed to "gas-particle partitioning" throughout the manuscript.

*2. Introduction, Paragraph 1: The authors are encouraged to also include mercury in the motivation to study atmospheric bromine chemistry.*
Reply: Thank you for the suggestion. We have included mercury depletion in the introduction.
Line 98-101:
"Halogen radicals, in particular the atomic chlorine (Cl·) and bromine (Br·), can deplete the $O_3$, react rapidly with VOCs with reaction rates up to two orders of magnitude faster than that of the hydroxyl radical (OH) reaction with VOCs and accelerate the depletion of gaseous elemental mercury (Atkinson et al., 2007;Calvert and Lindberg, 2004)."

*3. Lines 142-144: Additional inland measurements of chlorine chemistry include Mielke et al. 2016 (Can. J. Chem.) and McNamara et al. 2020 (ACS Central Sci.).*
Reply: Those two studies have been added to the main text.

Revised text in the main text:

line 141-142: "During the wintertime, the use of road salt could also be a dominant source of atmospheric Cl in the city areas (McNamara et al., 2020)."

line 142-145: "Follow-up studies have confirmed the presence of halogen activation spreading over the continental regions of North America, Canada, Europe and Asia (Mielke et al., 2011;Phillips et al., 2012;Riedel et al., 2013;Tham et al., 2016;Wang et al., 2017;Tham et al., 2018;Liu et al., 2017;Xia et al., 2020;Zhou et al., 2018;McNamara et al., 2020)."

*4. Lines 164-165: Please clarify this sentence as it not clear what species are included here, and the manuscript title mentions the stratosphere.*

Reply: Thank you for your comment. We noticed that is a typo and has been corrected.

Revised text in the main text (line 165-166):

"The atmospheric bromine is much less abundant than chlorine in the stratosphere with the concentrations of around 25 ppt (parts per trillion by volume) compared to 3.7 ppb of chlorine (Bedjanian and Poulet, 2003)".

*5. Lines 178-181 and 312: Note that McNamara et al. 2020 (ACS Central Sci.) also reported inland HCl measurements.*

Reply: This study has been involved and introduced in the main text.

Revised text in the main text (line 179-184):

"Some limited studies focused on the atmospheric HCl, for example, Crisp et al. (2014) summarized that the concentration of HCl is typically less than 1 ppb over the continental regions and McNamara et al. (2020) measured the concentration of HCl is around 100 ppt from inland sources, while an airborne measurement showed HCl concentrations of around 100 ppt was typically observed over the land area of northeast United States, except near power plant plumes with concentrations over 1 ppb (Crisp et al., 2014;McNamara et al. 2020;Haskins et al., 2018)."

Revised text in the main text (line 341-344):

"Although it is well known that the HCl is abundant in the polluted coastal and inland regions, previous studies show that the typical HCl mixing ratios over the continental urban areas are less than 1 ppb (Crisp et al., 2014;Faxon and Allen, 2013;Le Breton et al., 2018;McNamara et al. 2020), which are similar to our observations at Beijing."

*6. Line 223: Do the authors mean 5 s averaging here, as TOFs operate with much higher time resolution than 5 s.*

Reply: TOFs can operate with much higher time resolution, even less than 1 second. Due to our L-TOF was deployed for continuous and long-term ambient measurements, the raw L-TOF data were recorded at 5 s time resolution. For ambient measurements, the 5 s time resolution is sufficient. The data points reported in our manuscript were hourly-averaged ones.

*7. Line 302: Please add explanation of Figure 4d to this paragraph to improve the clarity of the discussion here.*

Reply: We have added a few sentences to clarify Figure 4d further.

Revised text in the main text (line 314-326):

"From Figure 4d, it also can be found that elevated temperature and high abundance of $HNO_3$ which was indicated as $[NO_2]*[OH]$ could intensify the HCl releases from particulate chloride in the daytime from 08:00 to 17:00. The OH radical concentrations were calculated using $J_{NO2}$ and $J_{O1D}$ (Section S8). This phenomenon is consistent with our observation results above where the increase

of temperature and UVB could reinforce the formation of chemicals (e.g., $HNO_3$) that promote the gas-particle partitioning or directly increase gas-phase formation rate of HCl and HBr (Crisp et al., 2014;Riedel et al., 2012), thus further enhancing the HCl and HBr (Figure 3). Although there is no direct measurement of particulate bromide (Br), considering the similarity in diurnal patterns and good correlation ($r$ = 0.70) between HBr and HCl (Figure 4c), and HBr tracking well with the temperature and $[NO_2]*[OH]$ (see Figure 3), it is rational to suppose HBr also predominantly derived from gas-particle partitioning process. The contribution by the reaction of bromine atoms with hydrocarbons to form HBr is likely not the dominant pathway as bromine atom is less reactive to hydrocarbons compared to the chlorine atom, and most often reacts with ozone (Simpson et al., 2015)."

*8. Figure 4: Please consider showing Parts a and b on linear scales in ppt. Part d needs more explanation in the caption. Is this a mass or mole ratio? Also, I did not see measurements of NO2 in the methods section, so are both NO2 and OH calculated values here? Please clarify here in the caption, on Line 374, and SI Line 275 how NO2 was obtained.*

Reply: Thanks for the suggestion. Yet, considering the consistency of the unit of manucript and also the suggestions on the units proposed by Reviewer 1#, in Figure 4 in the main text, we still use the unit as "molecules cm$^{-3}$" but on a linear scale (shown in Figure R6 in Marjor comment 6). We also added the following Figure R9 (Figure S12 in the revised SI) on linear scales in ppt as suggested to help the reader to compare with the studies using the unit of ppt. Figure capations of pandel d is also revised as suggested, which clear stated a mass ratio of HCl(g)/Cl(p) is presented.

[Figure]

**Figure R9 (Figure S12 in the revised SI).** Diurnal variations of UVB intensities, HCl and HBr concentrations (averaged values ± one standard deviation) (**a** and **b**) and the correlation between HCl and HBr (**c**). In panel c, the data points are hourly averaged ones during daytime (8:00-17:00). Temperature dependence of gas to particle partitioning ratios of mass concentration of chloride, colour-coded by [NO$_2$]*[OH] which was indicated as the abundance of HNO$_3$ (**d**). All snowy and rainy days during the sampling period were excluded.

The description of NO$_2$ measurement "NO$_2$ was measured with a THERMO 42i NO-NO$_2$-NO$_X$ Analyzer (Thermal Environment Instruments Inc. USA)" has been added to revised text in line 285-286 and SI (Section S1) line 89-90.

*9. Lines 364-366: Please clarify where these measurements were made, as this context is needed for understanding the relevance of the statement.*

Reply: Thanks for the comment. We have revised the text and added measurement information as suggested.

Revised text (line 395-400):

"It is also interesting to note that in a previous marine study conducted at Oahu, Hawaii, gaseous Br was found to be 4 to 10 times higher than particulate Br (Moyers and Duce, 1972). On the other hand, from a previous observation conducted in urban Beijing, high levels of both gaseous (7 ng m$^{-3}$) and particulate (in total suspended particles (TSP), 18 ng m$^{-3}$) bromine were measured by

offline sampling-organic solvent extraction and Instrumental Neutron Activation Analysis (INAA) method (Tian et al., 2005)."

*10. Lines 366-368: Please clarify these sentences. What is meant by extractable gaseous organic bromine? What is the statement "high concentration and reactivity of both organic/inorganic Br" referring to here (as only HBr is presented in this work and organic Br is generally not very reactive)?*

Reply: The original expression comes from the reference Tian et al. (2005), which applied Polyurethane Foam Infill (PUF) material to collect the atmospheric gaseous Br and then used organic solvent to extract. The organic solvent-extracted Br were finally analyzed by instrumental neutron activation analysis (INAA) method. Thus, the expression of "extractable gaseous organic bromine" represented the extracted organic Br collected from the offline gas-phase collection. Since the study is not closely related to the HBr measurement in our study, and to avoid possible misleading and confusions, we revised the aforementioned expressions.

Revised text (Line 395-400):

"It is also interesting to note that in a previous marine study conducted at Oahu, Hawaii, gaseous Br was found to be 4 to 10 times higher than particulate Br (Moyers and Duce, 1972). On the other hand, from a previous observation conducted in urban Beijing, high levels of both gaseous (7 ng $m^{-3}$) and particulate (in total suspended particles (TSP), 18 ng $m^{-3}$) bromine were measured by offline sampling-organic solvent extraction and Instrumental Neutron Activation Analysis (INAA) method (Tian et al., 2005)."

*11. Lines 375-378: Note that McNamara et al 2020 (ACS Central Sci) also reported similar Cl atom production rates from OH + HCl for the inland urban environment.*

Reply: This study has been included in the discussion.

Revised text in the main text (line 409-410):

"These rates fall within the range of Cl atom production rates ($\sim 10^3$ to $10^6$ molecules $cm^{-3}$ $s^{-1}$) reported in polluted environments (Crisp et al., 2014;Hoffmann et al., 2018;McNamara et al 2020)."

*12. Lines 382-383: Please provide references for this statement of previous work and also compare the Cl atom production rates from HCl in this work to ClNO2 and Cl2 photolysis in the previous studies.*

Reply: Agree. The text has been revised as the following (line 420- 425):

"Recent studies in several polluted sites of China suggested that the photolysis of $ClNO_2$ and $Cl_2$ are the dominant daytime Cl atom sources leading to Cl atom production rate up to $8 \times 10^6$ molecules $cm^{-3}$ $s^{-1}$ (Tham et al., 2016;Liu et al., 2017;Xia et al., 2020), while our observation of Cl atom production rate from HCl + OH is about $2 \times 10^3$ molecules $cm^{-3}$ $s^{-1}$. Despite the lower production rate, the reaction of HCl with OH may also act as important recycling of Cl atom, which ultimately enhanced the atmospheric oxidation capacity (Riedel et al.,2012)."

*13. Line 387: This statement about the ubiquity of bromine chemistry in the polluted urban*

*environment is speculative and not supported by the paper cited.*

Reply: We agree with the reviewer that the ubiquity of bromine chemistry in the polluted urban environment is speculative because there is no previous study that reported such species in the urban environment. However, a recent study found that $Br_2$ and BrCl are ubiquitous in the polluted environment of China which is influenced by biomass burning (Peng et al., 2020). This result together with the results from Lee et al. (2018) paper point to the fact that both biomass burning and coal-fired power plant are the major sources of bromine and these sources are quite widespread in the polluted environment of China. Therefore, we believe that bromine chemistry is ubiquitous in a polluted urban environment. We have tone down the statement and the revised sentence is as the following.

Line 425-429:

"In analogous to the chlorine chemistry, the reaction of HBr with OH could be a significant source of Br atom in the daytime although rapid photolysis of $Br_2$ and $BrNO_2$ is believed to be the major Br atom source in a polluted urban environment as ubiquitous bromine species (e.g. $Br_2$, BrCl and $BrNO_2$) have been previously observed in residential coal burning and coal-fired power plant plumes (Lee et al., 2018;Peng et al., 2021)."

*14. Lines 499-503: This reference is listed twice.*

Reply: The reference list has been corrected.

*15. Figure 1: The font on these maps is not readable as currently presented.*

Reply: Thanks for the suggestions and Figure 1 has already been changed accordingly.

[Figure]

**Figure R10 (Figure 1 in revised main text).** The location of BUCT measurement station. The satellite map was revised from © Google map.

*16. Figure 2: Consider showing HCl on the left axis and HBr on the right axis, both on linear scales. Incorporate measurement uncertainty.*

Reply: Figure 2 has already been changed accordingly and the updated Figure 2 was also attached here.

[Figure]

**Figure R11 (Figure 2 in revised main text).** The calculated enthalpy of HCl·NO$_2^-$ formed by HCl with NO$_2^-$ and Cl$^-$ with HNO$_2$ and enthalpy of HBr·NO$_2^-$ formed by HBr with NO$_2^-$ and Br$^-$ with HNO$_2$ at the DLPNO-CCSD(T)/def2-QZVPP// ωB97X-D/aug-cc-pVTZ-PP level of theory.

*17. Figure 7: Please clarify in the caption that all of the 'data' shown are calculations, not measurements, as implied by the text in the caption.*

Reply: The figure and its caption were revised accordingly. The revised caption:

"Time series of calculated production rates of Cl and Br radicals during the observation period (**a**); diurnal variations of HCl and HBr concentrations in clean and polluted days (**b**); diurnal variations of production rates of Cl and Br radicals, together with calculated OH radical concentrations (**c**) and production rates of Cl and Br radicals in clean and polluted days (**d**). The clean and polluted days were classified as daily PM$_{2.5}$ < 75 µg/m$^3$ and PM$_{2.5}$ ≥ 75 µg/m$^3$, respectively. The data points are in the hourly-average interval and measured during observation periods from 1 February to 31 March 2019."

---

## Author Response (AR2)

**RE:A point-to-point response to reviewers' comments**

Referee comments are given in italic blank and the changes made to the manuscript are given in blue. The line numbers correspond to the revised new manuscript.

*Reviewer #2*

*Fan et al present a revision of their manuscript reporting HCl and HBr observations in Beijing. The revisions made in response to reviewer comments significantly strengthened the manuscript. In particular, the reporting of uncertainties and LODs is important and useful. The move of now Figure 5 to the main text and associated caption and discussion edits are also excellent revisions that strengthen the manuscript. Similarly, the revisions to Figure 8 and added comparison of HCl and HBr on "clean" and polluted days are excellent.*

Reply: We are very grateful for the positive comments. And we have carefully revised our manuscript accordingly. A point-to-point response to reviewers' comments (in italic), is given below.

*I only have one remaining minor comment. It is not clear why the HCl and HBr concentrations represent lower limits (Line 257 of tracked changes version), as the reasoning for this is stated to be due to unquantified uncertainties such as different sensitivities to HCl and HBr, which reflect understanding of the value but not the directionality (lower/upper).*

Reply: Thanks for the reviewer's suggestion. In the original expression, we would like to express that the HBr concentrations should be treated as semi-quantified ones regarding to the charging of reagent ions. The reason is as follows:

Ideally, the concentration of HBr could be quantified by the HBr calibration coefficient multiplying the normalized $Br^-$ signals by total reagent ions (i.e., $NO_2^-$, $O_2^-$ and $NO_3^-$), which is shown in equation 1 (E1):

$$[HBr] = C_{HBr} \times \frac{(Br^-)}{(NO_2^-) + (O_2^-) + (NO_3^-)} \qquad \text{E (1)}$$

where [HBr] is the concentration of HBr and $C_{HBr}$ is the calibration coefficient of HBr. $(Br^-)$, $(NO_2^-)$, $(O_2^-)$, $(NO_3^-)$ represent the signals of $Br^-$, $NO_2^-$, $O_2^-$ and $NO_3^-$ from CI-APi-LTOF, respectively.

Since the HBr calibration coefficient was absent, we applied the calibration coefficient of HCl ($C_{HCl}$) achieved from the inter-comparison between MARGA and CI-APi-LTOF as an alternative of $C_{HBr}$ to semi-quantify HBr (E2).

$$[HBr] = C_{HCl} \times \frac{(Br^-)}{(NO_2^-) + (O_2^-) + (NO_3^-)} \qquad \text{E (2)}$$

However, considering the fraction of the cluster of HBr·NO$_3^-$ (or Br$^-$·HNO$_3$) to total Br- was less than 4%, the reaction pathway of HBr with NO$_3^-$ was not considered. Therefore, the HBr was quantified by the equation 3 (E3):

$$[HBr] = C_{HCl} \times \frac{(Br^-)}{(NO_2^-) + (O_2^-)} \qquad \text{E (3)}$$

Thus, the presented HBr concentrations should be treated as semi-quantified ones.

We have revised the statements in the manuscript accordingly to minimize the potential misunderstanding (line number correspond to revised MS):

**Line 251-254**: "Similar to HCl, the same uncertainty was also adopted for HBr mixing ratios. It should be noted that our assumptions lead towards a lower limit estimate of HCl and HBr concentrations, due to other potential uncertainties (e.g., different sensitivities of HCl and HBr) were not taken into account." has been revised. We also correct the typo in Line 254. The revised statement is as follows:

"Similar to HCl, the same uncertainty was also adopted for HBr mixing ratios. It should be noted that our assumptions lead towards a semi quantitative estimation of HBr concentrations, due to other potential uncertainties (e.g., different sensitivities of HCl and HBr) were not taken into account."

**Line 268-269**: "The presented HBr concentrations should be treated as semi-quantification ones and upper limit values." has been revised to "The presented HBr concentrations should be treated as semi-quantitative ones".

*Reviewer #3*

*This is an interesting study presenting new measurements of HCl and HBr in Beijing. I only have a few minor comments that the authors should consider addressing.*

Reply: We are very grateful for the positive comments and have carefully revised our manuscript accordingly.

*1. Use of [NO₂][OH] as proxy for HNO₃*

*Line 316. "HNO₃ which was indicated as [NO₂]*[OH]" I don't understand this statement. Are the authors measuring NO₂, calculating OH, and then assume that the product of the two is proportional to HNO₃? This doesn't seem correct. NO₂+OH is certainly one pathway for HNO₃ production, the other being heterogeneous reaction of N₂O₅ on aerosols, which is likely quite important in winter. Furthermore, the lifetime of HNO₃ is much longer than that of NO₂ or OH, so the product of [NO₂][OH] might not be a good proxy for the actual HNO₃ concentrations. It might be more useful to plot NO₂ and OH separately on figure 3.*

Reply: We agree with the reviewer that the reaction of $NO_2$ and OH is not the only pathway for $HNO_3$. The concentrations of $NO_2$ and OH were based on direct measurement and calculation, respectively (Section S8 in SI). However, $NO_2$+OH reaction would be one of the dominant pathways of $HNO_3$ during the daytime when elevated HCl and HBr were observed. In a previous study, this reaction is also regarded to be the largest sink of $NO_x$ globally (Stavrakou et al., 2013). The formation pathways of gaseous $HNO_3$ from the heterogeneous reactions of $N_2O_5$ would be much more important during the night time, especially during the haze periods (Wang et al., 2020).

We did notice that the lifetime of $HNO_3$ could be longer than $NO_2$ or OH (Amedro et al., 2020; Hanke et al., 2003), which may result in uncertainties in $HNO_3$ estimations. Considering those factors mentioned above, we toned down the expression of $NO_2$, OH and their product in the main text and separately exhibit [$NO_2$] and [OH] in Figure 3 as suggested.

To make our statement clear, we revised the manuscript accordingly:

**Line 316-321**:

From: "Figure 4d, it also can be found that elevated temperature and high abundance of $HNO_3$ which was indicated as [$NO_2$]*[OH] could intensify the HCl releases from particulate chloride in the daytime from 08:00 to 17:00."

To: "From Figure 4d, it also can be found that elevated HCl is associated with high temperature and [$NO_2$]*[OH] value. Considering the reaction of $NO_2$ with OH radical is one of the dominant formation pathways of gaseous $HNO_3$ during the daytime (Stavrakou et al., 2013), it implies that strong photochemical reactions and the following potential elevated $HNO_3$ could intensify

the HCl releases from particulate chloride in the daytime from 08:00 to 17:00."

According to the suggestions, we revised Figure 3 (Figure R1) as shown.

[Figure]

**Figure R1 (Figure 3 in main text).** Time profiles of temperature (**a**), UVB intensities (**b**), $NO_2$ concentration (**c**), OH concentration from calculation (**d**), $[NO_2]*[OH]$ (µg m$^{-3}$ * molecules cm$^{-3}$) (**e**), particulate chloride concentration (Cl(p)) (**f**) and the mixing ratios of HCl and HBr (**g**). The data points are in hourly-average interval.

*2. OH calculation*

*Line 315 and supplemental info. The method and justification for calculating OH is not very clear. It seems that it is based on $J_{O1D}$ and $NO_2$ measurements and based on inferred $JNO_2$. S8 cites 2 papers: Xu et al. and Tan et al. It is unclear whether the empirical fit from Xu et al. (2015) in equation S2 was originally obtained from observations of OH concentrations or model calculations. Also, the authors then go on to say (SI, lines 359-365), that they use the results from Tan et al. (2018), which they mention is only a function of $J_{O1D}$. Can the authors clarify the whole section? What equation do they use (the one from Xu or from Tan)? What is that equation based on? Pure model calculations (and if so, what are the main sources of OH?) or actual observations of OH?*

Reply: Thanks for reviewer's comment. In this study, we applied the empirical equation ([OH] = $J_{O1D} \times 2 \times 10^{11}$) from Tan et al., 2019 to estimate the OH concentrations, which is proposed by

the direct OH measurement by LIF (Laser-Induced Fluorescence) in the North China Plain region. We then applied a more detailed calculation proposed by Xu et al., 2015 to validate the calculation. Comparable results were achieved from these two methods in both concentration levels and diel patterns (as shown in Figure S11). There is a typo in the text (Line 363), the citation should be "Tan et al., (2019)" and has been corrected through the entire SI.

To make the OH calculation method clear, we revised the supplemental accordingly (line number correspond to revised SI):

**Line 355-369**:

"In this work, there is no direct measurement of OH radical concentration during observation periods. While during the winter and spring in Beijing, it has been found that the measured OH radical concentration is linearly correlated with photolysis rate of ozone, $J_{O1D}$ (Liu et al., 2020; Tan et al., 2019). Thus, an empirical equation was proposed to estimate the OH concentrations: [OH] = $J_{O1D}$×2×10$^{11}$ molecules cm$^{-3}$. We adopted this empirical equation to calculate the OH concentration in this study.

We further validated our calculation by comparing the OH concentration, obtained with another method suggested by Xu et al. 2015 (Xu et al., 2015), which considering both photolysis rate ($J_{O1D}$ and $J_{NO_2}$) and NO$_2$ concentration ($C_{NO_2}$) based on formula equation (S2). Using another dataset collected from 21 May to 10 June 2019 where the parameters of $J_{NO_2}$ and $C_{NO2}$ were available from direct measurements, a good correlation (r=0.97) was achieved between measured $J_{NO_2}$ and predicted $J_{NO_2}$ which was derived from the solar zenith angle and the location using a box model (FACSIMILE 4) (Liu et al., 2020), confirming the validation of our predicted $J_{NO_2}$ (Figure S11a)."

$$c_{OH} = \frac{4.1\times10^9\times(J_{O1D})^{0.83}\times(J_{NO_2})^{0.19}\times(140c_{NO_2}+1)}{0.41c_{NO_2}^2+1.7c_{NO_2}+1} \qquad \text{Eq. (S2)}$$

[Figure]

**Figure R2 (Figure S11).** High correlation ($r = 0.97$) between measured and predicted $J_{NO_2}$ from 21 May to 10 June 2019 (**a**); Calculated diurnal curve of OH concentration based on Tan et al., (2019) and Xu et al., (2015) from 1 February to 31 March 2019 (**b**).

***Additional Comments:***

*Page 4 line 187: "is of necessary" should be replaced with "is necessary"*

Reply: "is of necessary" has been changed to "is necessary" in the manuscript.

*Page 6 260 "due to a direct calibration for HBr" should be replaced with "as direct calibration for HBr"*

Reply: It has been revised accordingly.

*Line 268 "semi-quantification" should be replaced with "semi quantitative" as direct calibration for HBr*

Reply: It has been changed accordingly throughout the manuscript.

*Figure 5. The figure is difficult to read with different units plotted molec/cm³, mol/mol, ug/m³. I suggest that the authors stick to one unit: pptv. Given that the point of the figure is to look at*

*the ratio between HCl and pCl this will make the figure more straightforward to read. Also, the panel on the right has 2 y axis with Cl(p) and none with HCl.*

Reply: Thank you for the suggestion. We agree that it is difficult to read with different units in the plot. By considering the consistency of the unit of manuscript in Figure 4 of the main text, we prefer to use the unit as "molecules cm$^{-3}$" for concentration of gas-phase species and "μg m$^{-3}$" for concentration of particulate compounds. Meanwhile, we also added the following Figure R3 in pptv to the SI (Figure S13 in the revised SI) as suggested by the reviewer to help the readers to compare with previous studies using the unit of pptv.

[Figure]

**Figure R3 (Figure S13 in SI).** Time variation of daily averaged concentration of particulate chloride (Cl(p)) measured by ACSM, gaseous HCl (HCl(g)) measured by CI-APi-LTOF and mole ratios of HCl(g)/Cl(p) (a) and diurnal variation of HCl(g), Cl (p) and mole ratios of HCl(g)/Cl(p) (b). Note that the plots are similar to those in Figure 5 of the main text, but this is displayed as pptv.

The typo in Figure 5 panel b on the left y-axis has been corrected to "HCl (g) Conc. (molec. cm$^{-3}$)" shown as Figure R4.

[Figure]

**Figure R4 (Figure 5 in main text).** Time variation of daily averaged concentration of particulate chloride (Cl(p)) measured by ACSM, gaseous HCl (HCl(g)) measured by CI-APi-LTOF and mole ratios of HCl(g)/Cl(p) (**a**) and diurnal variation of HCl(g), Cl (p) and mole ratios of HCl(g)/Cl(p) (**b**).

*Text unclear as to whether the HBr values reported are upper or lower limits. Line 252 "lower limit estimates" while line 268 states "HBr concentrations should be treated as semi-quantification ones and upper limit values".*

Reply: Thank you for the suggestion. Please refer to our reply for the comment from reviewer#2. For the convenience, the reply was copied below:

In the original expression, we would like to express that the HBr concentrations should be treated as semi-quantified ones regarding to the charging of reagent ions. The reason is as follows:

Ideally, the concentration of HBr could be quantified by the HBr calibration coefficient multiplying the normalized $Br^-$ signals by total reagent ions (i.e., $NO_2^-$, $O_2^-$ and $NO_3^-$), which is shown in equation 1 (E1):

$$[HBr] = C_{HBr} \times \frac{(Br^-)}{(NO_2^-) + (O_2^-) + (NO_3^-)} \qquad \text{E (1)}$$

where [HBr] is the concentration of HBr and $C_{HBr}$ is the calibration coefficient of HBr. $(Br^-)$, $(NO_2^-)$, $(O_2^-)$, $(NO_3^-)$ represent the signals of $Br^-$, $NO_2^-$, $O_2^-$ and $NO_3^-$ from CI-APi-TOF, respectively.

However, since the HBr calibration coefficient was absent, we applied the calibration coefficient of HCl ($C_{HCl}$) achieved from the intercomparison between MARGA and CI-APi-LTOF as an alternative of $C_{HBr}$ to semi-quantify HBr (E2).

$$[HBr] = C_{HCl} \times \frac{(Br^-)}{(NO_2^-) + (O_2^-) + (NO_3^-)} \qquad \text{E (2)}$$

However, considering the fraction of the cluster of $HBr \cdot NO_3^-$ (or $Br^- \cdot HNO_3$ ) to total Br- was less than 4%, the reaction pathway of HBr with $NO_3^-$ was not considered. Therefore, the HBr was quantified by the equation 3 (E3):

$$[HBr] = C_{HCl} \times \frac{(Br^-)}{(NO_2^-) + (O_2^-)} \qquad \text{E (3)}$$

Thus, the presented HBr concentrations should be treated as semi-quantified ones.

We have revised the statements in the manuscript accordingly to minimize the potential misunderstanding (line number correspond to revised MS):

**Line 250-253**: "Similar to HCl, the same uncertainty was also adopted for HBr mixing ratios. It should be noted that our assumptions lead towards a lower limit estimate of HCl and HBr concentrations, due to other potential uncertainties (e.g., different sensitivities of HCl and HBr) were not taken into account." has been revised. We also correct the typo in Line 254. The revised statement is as follows:

"Similar to HCl, the same uncertainty was also adopted for HBr mixing ratios. It should be

noted that our assumptions lead towards a semi-quantitative estimation of HBr concentrations, due to other potential uncertainties (e.g., different sensitivities of HCl and HBr) were not taken into account."

**Line 267-268**: "The presented HBr concentrations should be treated as semi-quantification ones and upper limit values." has been revised to "The presented HBr concentrations should be treated as semi-quantitative ones".

**References:**

Amedro, D., Berasategui, M., Bunkan, A. J. C., Pozzer, A., Lelieveld, J., and Crowley, J. N.: Kinetics of the $OH + NO_2$ reaction: effect of water vapour and new parameterization for global modelling, Atmospheric Chemistry and Physics, 20, 3091-3105, 10.5194/acp-20-3091-2020, 2020.

Hanke, M., Umann, B., Uecker, J., Arnold, F., and Bunz, H.: Atmospheric measurements of gas-phase $HNO_3$ and $SO_2$ using chemical ionization mass spectrometry during the MINATROC field campaign 2000 on Monte Cimone, Atmos. Chem. Phys., 3, 417-436, 10.5194/acp-3-417-2003, 2003.

Liu, Y., Zhang, Y., Lian, C., Yan, C., Feng, Z., Zheng, F., Fan, X., Chen, Y., Wang, W., Chu, B., Wang, Y., Cai, J., Du, W., Daellenbach, K. R., Kangasluoma, J., Bianchi, F., Kujansuu, J., Petäjä, T., Wang, X., Hu, B., Wang, Y., Ge, M., He, H., and Kulmala, M.: The promotion effect of nitrous acid on aerosol formation in wintertime in Beijing: the possible contribution of traffic-related emissions, Atmos. Chem. Phys., 20, 13023-13040, 10.5194/acp-20-13023-2020, 2020.

Stavrakou, T., Müller, J. F., Boersma, K. F., van der A, R. J., Kurokawa, J., Ohara, T., and Zhang, Q.: Key chemical NOx sink uncertainties and how they influence top-down emissions of nitrogen oxides, Atmospheric Chemistry and Physics, 13, 9057-9082, 10.5194/acp-13-9057-2013, 2013.

Tan, Z., Lu, K., Jiang, M., Su, R., Zhang, Y.: Daytime atmospheric oxidation capacity in four Chinese megacities during the photochemically polluted season: A case study based on box model simulation. Atmospheric Chemistry and Physics 19, 3493-3513, 2019.

Wang, Y., Chen, Y., Wu, Z., Shang, D., Bian, Y., Du, Z., Schmitt, S. H., Su, R., Gkatzelis, G. I., Schlag, P., Hohaus, T., Voliotis, A., Lu, K., Zeng, L., Zhao, C., Alfarra, M. R., McFiggans, G., Wiedensohler, A., Kiendler-Scharr, A., Zhang, Y., and Hu, M.: Mutual promotion between aerosol particle liquid water and particulate nitrate enhancement leads to severe nitrate-dominated particulate matter pollution and low visibility, Atmospheric Chemistry and Physics, 20, 2161-2175, 10.5194/acp-20-2161-2020, 2020.

Xu, Z., Wang, T., Wu, J., Xue, L., Chan, J., Zha, Q., Zhou, S., Louie, P.K.K., Luk, C.W.Y.: Nitrous acid (HONO) in a polluted subtropical atmosphere: Seasonal variability, direct vehicle emissions and heterogeneous production at ground surface. Atmospheric Environment 106, 100-109, 2015.